# STRATEGY-DRIVEN CENTRAL LIMIT THEOREM FOR SEQUENTIAL TEST

## ABSTRACT

A/B testing is a critical tool for evaluating the effectiveness of strategies, but its conclusions are typically limited to the Average Treatment Effect (ATE). However, a more fundamental question arises when deciding whether to implement personalized interventions: whether Heterogeneous Treatment Effects (HTE) exist. This paper addresses the challenge of testing for the existence of HTE. While current methods based on the t-test are effective, the core pursuit of statistical inference is to enhance test power to more sensitively detect subtle heterogeneous effects. To this end, this paper proposes a novel sequential testing framework based on Strategy Limit Theory, specifically designed to more effectively identify these hard-to-detect, subtle differences. The main contributions are as follows: (i) We integrate HTE existence testing into a strategic decision-making process and construct a new test statistic based on Strategy Limit Theory, weighted by parameter $\lambda$ to control Type I error. By maximizing the divergence between the distributions under the null and alternative hypotheses, we enhance the test's power. (ii) We extend this approach to online experimental settings and introduce a Bi-Optimal Strategy (BOS). This strategy not only improves statistical power but also significantly enhances the cumulative reward of the experiment. (iii) We develop a complete sequential testing procedure. By combining the alpha-spending function with the Bootstrap method, we determine dynamic stopping boundaries to accommodate the complex joint distribution of our statistic. (iv) We validate the effectiveness and superiority of our proposed method through extensive simulation experiments and empirical analysis on Tenrec, a real-world dataset from Tencent's recommendation system.

## 1 INTRODUCTION

A/B testing, also known as randomized controlled trials, is the gold standard for evaluating the effectiveness of new strategies, products, and interventions. Widely used in both industry and academia, its core conclusion typically focuses on the Average Treatment Effect (ATE), which measures the average impact of an intervention across all individuals (Lai, 2001). However, the ATE's limitation is that it can obscure individual-level differences. In real-world scenarios, a strategy may have a positive effect on some groups while having a limited or even negative effect on others. This phenomenon, where the treatment effect varies with individual characteristics, is known as Heterogeneous Treatment Effects (HTE). This is a topic of significant interest in fields like social science, economics, and medicine (Xie et al., 2012; Vivalt, 2015; Dahabreh et al., 2016).

Understanding and leveraging HTE is critical in areas such as personalized medicine, precision marketing, and adaptive education. The premise for all personalized strategies is the existence of HTE. If the treatment effect were constant for all individuals, a single, universal strategy would be optimal, and there would be no need for personalized interventions (Grimmer et al., 2017). Therefore, before investing resources into developing complex and computationally intensive personalized models (e.g., causal forests, metalearners, and deep learning models) (Wager & Athey, 2018; Künzel et al., 2019; Curth & Van der Schaar, 2021), a more fundamental and critical question is: Is there sufficient statistical evidence to prove the existence of HTE? That is, we need a rigorous hypothesis test for the existence of HTE.

Existing literature has proposed several methods for testing the existence of HTE. Classic approaches often focus on specific HTE patterns, such as testing for qualitative interactions, where an intervention produces effects in opposite directions across different subgroups (Gail & Simon, 1985; Roth & Simon, 2018). To adapt to the dynamic nature of online data streams, recent research has extended this to a sequential testing framework (Shi et al., 2021). While these methods are effective, a core goal of statistical inference is to further enhance statistical power to more sensitively detect subtle heterogeneous effects (Lai, 2001). Ignoring weak HTE signals could lead to missing significant opportunities for personalized optimization.

To address these challenges, this paper proposes a novel sequential testing framework based on Strategic Limit Theory (Chen et al., 2022). The core idea is to transform the HTE existence test into a strategic decision-making process, structurally similar to a multi-armed bandit(Feldman, 1962; Berry, 1972; Vogel, 1960) but aimed at maximizing statistical power. By strategically collecting data, we construct a test statistic that maximizes the distributional divergence between the null (no HTE) and alternative (HTE exists) hypotheses(Chen et al., 2023). This amplification of the underlying signal enables more reliable detection of subtle heterogeneous effects that are often missed by traditional methods.

The main contributions of this paper are as follows:

- We propose an innovative strategic testing framework that formulates the HTE existence test as a dynamic decision process. We then design a new test statistic based on Strategy Limit Theory, weighted by $\lambda$ to balance mean and volatility terms and control Type I error, which aims to maximize the distributional divergence between the null and alternative hypotheses to enhance statistical power.

- We extend this framework to online experimental settings and introduce a Bi-Optimal Strategy (BOS). This strategy not only maintains high statistical power but also significantly improves the cumulative reward of the experiment, mitigating the ethical and cost concerns of traditional sequential testing.

- We develop a complete sequential testing procedure. By innovatively combining the alpha-spending function (Gordon Lan & DeMets, 1983) with the Bootstrap method, we precisely determine dynamic stopping boundaries.

- We validate the efficiency and superiority of our method in detecting HTE through large-scale simulation experiments and empirical analysis using Tenrec (Yuan et al., 2022), a real-world dataset from Tencent's recommendation system.

The rest of this paper is structured as follows: Section 2 provides background and problem definitions; Section 3 details our methodology, including the $\lambda$-weighted statistic and the incorporated Strategic Central Limit Theorem; Section 4 describes the sequential testing procedure; Sections 5 and 6 present the results of our simulation studies and case analysis, respectively; Section 7 concludes with a summary and discussion.

## 2 BACKGROUND AND PROBLEM STATEMENT

### 2.1 POTENTIAL OUTCOMES FRAMEWORK

To rigorously explore individual-level differences in treatment effects, our study is built upon the potential outcomes framework. We observe a sequential data stream of triplets $\{(X_j, A_j, Y_j)\}_{j=1}^{J}$. Here, $X_j \in \mathbb{R}^p$ is the covariate vector for the $j$-th observational unit, $A_j \in \{0,1\}$ is a binary variable representing the treatment assigned to the unit, and $Y_j$ is the observed outcome or reward. A larger value for $Y_j$ typically indicates a more desirable outcome. For each unit $j$, we define two potential outcomes: $Y_j^*(1)$ and $Y_j^*(0)$. $Y_j^*(1)$ is the outcome that unit $j$ would have if it received the treatment ($A_j = 1$), while $Y_j^*(0)$ is the outcome if it received the control ($A_j = 0$).

### 2.2 HYPOTHESIS TESTING FOR HETEROGENEOUS TREATMENT EFFECTS

Heterogeneous Treatment Effects (HTE) refer to the phenomenon where the effect of a treatment intervention varies depending on individual characteristics (i.e., covariates). If HTE does not exist,

it implies that the treatment effect is constant for all individuals, and a single, unified strategy is optimal. Conversely, the existence of HTE provides the theoretical basis for personalized interventions. Our primary objective is to test for the existence of HTE. To do so, we focus on whether the treatment effect for a subgroup with specific covariates $x$, given by $\tau(x) = \mathbb{E}[Y^*(1) - Y^*(0)|X = x]$, is consistently zero. In practice, directly solving for the conditional expectation $\mathbb{E}[Y^*(a)|X = x]$ is challenging. Thus, we introduce a basis function $\varphi(x)$ to approximate the conditional expected reward $Q_0(x, a) = \mathbb{E}[Y^*(a)|X = x]$. Specifically, we set $Q_0(x, a) \approx \varphi(x)^\top \beta_a^*$, where $\beta_a^* \in \mathbb{R}^q$ is a parameter vector to be estimated. The treatment effect for a specific subgroup can then be approximated as:

$$\tau(x) \approx \varphi(x)^\top (\beta_1^* - \beta_0^*).$$

Based on this, the problem of testing for the existence of HTE can be formalized as a hypothesis test:

$$\mathbf{H_0} : \varphi^\top(x)\,(\beta_1^* - \beta_0^*) = 0, \forall x \in \mathcal{X} \quad \text{vs} \quad \mathbf{H_1} : \varphi^\top(x)\,(\beta_1^* - \beta_0^*) \neq 0, \exists x \in \mathcal{X}.$$

Here, $\mathcal{X}$ is the support set of the covariate $X$. Rejecting $\mathbf{H_0}$ indicates that a treatment effect exists, and this effect is non-zero in at least some subgroups.

## 3 METHODOLOGY

### 3.1 A STRATEGIC FRAMEWORK FOR HTE TESTING

To effectively test for the existence of HTE, this paper introduces a novel testing framework. This framework draws on the idea of strategic two-sample testing, framing the hypothesis testing problem as an abstract strategic decision-making process, structurally similar to a Two-Armed Bandit (TAB).

It is important to note that the TAB framework we construct is a statistical decision tool, not a direct online experiment or clinical trial design. The two "arms" in this framework, which we call Left (L) and Right (R), do not represent the actual treatment options (e.g., treatment group vs. control group). Instead, they represent two different strategies used at each step of the test to update the statistical evidence. At step $j$ of the testing process, we make a strategic decision $\vartheta_j \in \{0, 1\}$ (0 for choosing the L arm, 1 for the R arm) based on historical information. This decision is designed to construct an optimal update term $U_j^\theta(x)$. The ultimate goal of this sequence of strategic choices, $\theta = \{\vartheta_1, ..., \vartheta_J\}$, is to shape the probability distribution of the final test statistic so that it is maximally distinguishable under $\mathbf{H_0}$ and $\mathbf{H_1}$, thereby boosting the test's statistical power.

### 3.2 THE TEST STATISTIC AND ITS DISTRIBUTION

This subsection details the construction of our proposed test statistic. First, at step $j$, we construct $U_j^\vartheta(x)$ based on the strategic decision $\vartheta_j$:

$$U_j^\theta(x) = \begin{cases} W_j^L = \varphi^T(x)\{\hat{\Sigma}_{0,j}^{-1}\varphi(X_j)Y_j - \hat{\Sigma}_{1,j}^{-1}\varphi(X_{1,j}^*)Y_{1,j}^*\}, & \text{if } \vartheta_j = 0 \\ W_j^R = \varphi^T(x)\{\hat{\Sigma}_{1,j}^{-1}\varphi(X_j)Y_j - \hat{\Sigma}_{0,j}^{-1}\varphi(X_{0,j}^*)Y_{0,j}^*\}, & \text{if } \vartheta_j = 1 \end{cases}$$

where $\hat{\Sigma}_{a,j}$ is the estimated covariance matrix based on historical data up to step $j - 1$ for group $a$ ($a = 0$ or $1$). $(X_{a,j}^*, Y_{a,j}^*)$ represents an observation randomly sampled from the historical data of the same treatment group, used to construct the other part of the update term.

After a total of $J$ steps, we construct the final test statistic. Statistic is designed as a weighted combination of a mean term and a volatility term. The standard approach effectively gives these terms equal weighting. However, inspired by recent work in strategic A/B testing (Zhang et al., 2025), we introduce a weighting parameter $\lambda$, to more flexibly tune the test's properties. This parameter is crucial for balancing the trade-off between achieving greater statistical power and maintaining rigorous control over the Type I error rate. A larger value for $\lambda$ generally boosts statistical power but can slow the statistic's convergence and inflate the Type I error if chosen improperly. The resulting weighted statistic for any given covariate $x$ is formulated as:

$$S_{J,\lambda}(x, \theta) = \frac{1}{J} \sum_{j=1}^{J} \frac{\lambda}{1 - \lambda} U_j^\theta(x) + \frac{1}{\sqrt{J}} \sum_{j=1}^{J} \frac{U_j^\theta(x)}{\hat{\sigma}_J}, \tag{1}$$

where $\lambda \in (0,1)$ and $\hat{\sigma}_J$ is a consistent estimator of $\mathrm{var}(U_j^\theta(x))$.

To test for the global existence of HTE, we use its supremum over all $x \in \mathcal{X}$ as the final test statistic:

$$S_{J,\lambda}(\theta) = \sup_{x \in \mathcal{X}} S_{J,\lambda}(x, \theta), \tag{2}$$

According to Strategy Limit Theory (Chen et al., 2022), for a specific strategy $\theta$ to be introduced in subsequent sections, this statistic's asymptotic distribution under $\mathbf{H_0}$ is a standard normal distribution. However, under the alternative hypothesis $\mathbf{H_1}$, its asymptotic distribution will exhibit a unique "bi-normal distribution", with a probability density function of the form:

$$f^\kappa(y) = \frac{1}{\sqrt{2\pi}} e^{-\frac{(|y|-\kappa)^2}{2}} - \kappa e^{2\kappa|y|} \Phi(-|y| - \kappa).$$

As shown in Figure 1, this distribution is more "flattened" and bimodal than the standard normal distribution. It is this significant distributional divergence between $\mathbf{H_0}$ and $\mathbf{H_1}$ that allows our testing method to more sensitively capture the existence of treatment effects, thereby achieving higher statistical power.

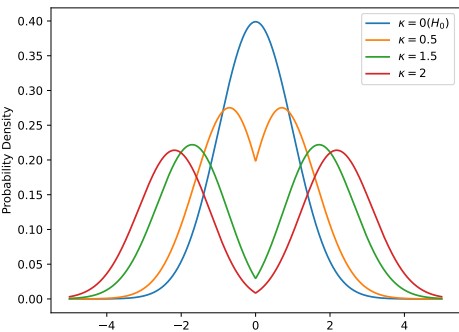

Figure 1: Density plots of bi-normal distribution across different $\kappa = 0.5, 1.5$, and $2$ vs. the density plot of standard normal distribution (blue line, as $\kappa = 0$).

### 3.3 Optimal Strategy

To maximize statistical power, this paper introduces an **Optimal Strategy (OS)**, denoted as $\theta^{os}$. The core idea of this strategy is to dynamically choose a sequence of actions that, when the alternative hypothesis $\mathbf{H_1}$ is true, maximizes the volatility of the test statistic, making it easier to reject the null hypothesis.

**Lemma 3.1.** *(Optimal strategy $\theta^{os}$) For any $0 \le l < \infty$, we can construct strategies $\theta^{os} = (\vartheta_1^{os}, \cdots, \vartheta_J^{os})$ as follows:*

$$\vartheta_j^{os} = \begin{cases} 0, & S_{j-1,\lambda}(x, \theta^{os}) > 0, \\ 1, & S_{j-1,\lambda}(x, \theta^{os}) \le 0, \end{cases} \quad \text{for } j \ge 1 \tag{3}$$

*such that*

$$\lim_{J \to \infty} \Pr\left(|S_{J,\lambda}(x, \theta^{os})| \ge l\right) = \lim_{J \to \infty} \sup_{\theta \in \Theta} \Pr\left(|S_{J,\lambda}(x, \theta^{os})| \ge l\right) \ge \Pr(|T| \ge l) \tag{4}$$

*where $T$ is the standard T-Test statistic that follows the normal distribution as a baseline method.*

The equality on the right side of Equation 4 holds if and only if $Q_0(x, 0) = Q_0(x, 1)$. The intuition behind this decision rule is that if the previous step's statistic $S_{j-1,\lambda}$ is positive, we choose an update strategy (arm L) that tends to make it negative in the current step, and vice versa. This alternating update mechanism ensures that when $\mathbf{H_0}$ is true, the statistic still converges to a standard normal distribution. However, when $\mathbf{H_1}$ is true, this mechanism significantly increases the statistic's volatility, causing it to asymptotically converge to the aforementioned bi-normal distribution. This makes it much easier to cross the rejection boundary, thus boosting the test's power.

## 3.4 BI-OPTIMAL STRATEGY

In offline testing, the test statistic constructed with the OS demonstrates powerful testing performance. However, in online testing scenarios (especially in medical trials), the cumulative reward during the experiment is a critical factor that must be considered. The OS, in its pursuit of maximizing test power, might choose a less effective intervention, raising ethical and safety concerns.

To address this, we propose a novel contribution: the **Bi-Optimal Strategy (BOS)**, denoted as $\theta^*$. This strategy is designed to achieve a dual objective: (i) maintain the high test power provided by the OS; and (ii) maximize the cumulative reward assigned to experimental units during the experiment.

**Theorem 3.2.** *(Bi-optimal strategy $\theta^*$) For any $0 \le l < \infty$, we can construct Bi-optimal strategies* $\theta^* = (\vartheta_1^*, \cdots, \vartheta_J^*)$ *as follows:*

$$
\vartheta_j^* = \begin{cases} 0, & \mathbb{I}\{S_{j-1,\lambda}(x, \theta^*)\}\, \mathbb{I}\left\{\widehat{Q}_0(x, 0) - \widehat{Q}_0(x, 1)\right\} \ge 0, \\ 1, & \mathbb{I}\{S_{j-1,\lambda}(x, \theta^*)\}\, \mathbb{I}\left\{\widehat{Q}_0(x, 0) - \widehat{Q}_0(x, 1)\right\} < 0. \end{cases} \tag{5}
$$

*Under the Bi-optimal strategy, the test statistics $S_{J,\lambda}(\theta^*)$ exhibit the following asymptotic properties. Let $\varphi \in C(\overline{\mathbb{R}})$ be a continuous function on $\mathbb{R}$ with finite limits at $\pm\infty$, and be an even function and monotone on $(0, \infty)$, then the limit distributions of $\{S_{J,\lambda}(\theta^*)\}$ satisfies $\lim_{J \to \infty} E_P[\varphi(S_{J,\lambda}(\theta^*))] = E_P[\varphi(\sigma_0\eta_J)]$, where $\eta_J \sim \mathcal{S}(m/\sigma_0, \Delta_J/\sigma_0)$ with the initial value $m$ and parameter $\Delta_J = \lambda|\Delta|/(1-\lambda) + \sqrt{J}|\Delta|/\sigma$, $\sigma_0 = \sqrt{1 + \Delta^2/\sigma^2}$. The density function of skewed binormal distribution is*

$$
f^{\frac{m}{\sigma_0}, \frac{\Delta_J}{\sigma_0}}(y) = \frac{1}{\sqrt{2\pi}} e^{-\frac{(y-m/\sigma_0)^2 - 2\Delta_J(|y|-|m|/\sigma_0)/\sigma_0 + \Delta_J^2/\sigma_0^2}{2}} - \frac{\Delta_J}{\sigma_0} e^{\frac{2\Delta_J|y|}{\sigma_0}} \Phi\left(-|y| - \frac{|m|}{\sigma_0} - \frac{\Delta_J}{\sigma_0}\right).
$$

This strategy combines the alternating update mechanism of the OS with a greedy approach: when deciding the form of the statistic, it also considers the current estimated rewards for each treatment option, $\hat{Q}_0(x, a)$, and tends to choose the option with the higher estimated reward. This design ensures that while maintaining high test power, the safety and effectiveness of the experimental process are improved. As shown in Figure 2a, under the Bi-optimal Strategy (BOS), the asymptotic distribution of the test statistic $S_{J,\lambda}(\theta^*)$ is compared to the strategy $\theta^{os}$ from Lemma 4. Under strategy $\theta^*$, the agent is more likely to choose a behavior with a higher reward. On the other hand, as shown in Figure 2b, under the alternative hypothesis, the asymptotic distribution of $S_{J,\lambda}(\theta^*)$ is still more dispersed than the classic normal distribution, which indicates that the bi-optimal strategy retains the stronger test power brought by the bi-normal distribution.

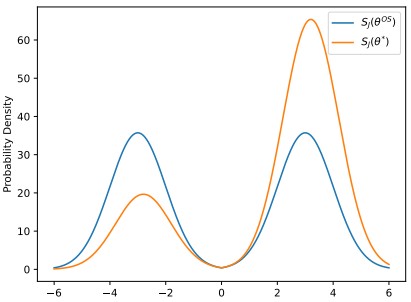

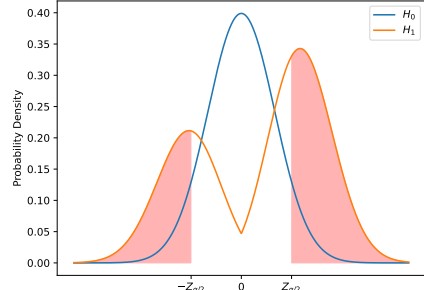

(a) Density plots of Test Statistics $S_{J,\lambda}(\theta)$ under different strategy: optimal strategy $\theta^{os}$ (blue line) and bi-optimal strategy $\theta^*$ (orange line).

(b) The shadow denotes the power of testing statistic $S_{J,\lambda}(\theta^*)$ i.e., $\Pr\left(|S_{J,\lambda}(\theta^*)| > z_{\alpha/2}|\mathrm{H}_1\right)$ under $d_0 = \sup_{x \in \mathcal{X}}(Q_0(x, 1) - Q_0(x, 0)) = 1$.

## 3.5 ASYMPTOTIC PROPERTIES AND POWER ANALYSIS

This section analyzes the power advantage of the BOS. Theorem 5 shows that under the BOS, when $\mathbf{H_0}$ is true, the statistic still converges to a standard normal distribution. However, when $\mathbf{H_1}$ is true,

the test statistic follows a skewed bi-normal distribution. Based on these findings, the following theorem demonstrates that for a sufficiently large sample size $J$, the test power $1 - \beta_2$ of our BOS-based method is greater than or equal to the power $1 - \beta_1$ of a traditional adaptive t-test.

**Theorem 3.3.** *(i) **Power Analysis:** The critical value $z_{\alpha/2}$ is calculated under the standard normal distribution and the rejection region corresponds to the light red area in Figure 2b. Let $d_0 = \sup_{x \in \mathcal{X}}(Q_0(x, 1) - Q_0(x, 0))$, the statistical power can be approximated by:*

$$1 - \beta_2 = \lim_{n \to \infty} \Pr(|S_{J,\lambda}(\theta)| > Z_{\alpha/2}|\mathbf{H_1}) = 1 - \Phi\left(-\frac{d_0 - z_{\alpha/2}}{\sigma_0}\right) + e^{\frac{2d_0 z_{\alpha/2}}{\sigma_0^2}} \Phi\left(-\frac{d_0 + z_{\alpha/2}}{\sigma_0}\right) \tag{6}$$

*For a sufficiently large J, we can obtain*

$$1 - \beta_2 \geq 1 - \Phi\left(\frac{|d_0| + \frac{\sqrt{J}|d_0|}{\sigma} + z_{\alpha/2}}{\sigma_0}\right) + \Phi\left(\frac{|d_0| + \frac{\sqrt{J}|d_0|}{\sigma} - z_{\alpha/2}}{\sigma_0}\right)$$

$$\geq 1 - \Phi\left(\frac{\sqrt{J/2}|d_0|}{\sigma} + z_{\alpha/2}\right) + \Phi\left(\frac{\sqrt{J/2}|d_0|}{\sigma} - z_{\alpha/2}\right) \quad (s.t.; J \geq \sqrt{\sigma}|d_0|)$$

$$= 1 - \beta_1. \tag{7}$$

*(ii) **Type I error control:** Under $\mathbf{H_0}$, the test statistic $S_{J,\lambda}(\theta)$ satisfies*

$$\lim_{n \to \infty} \Pr(|S_{J,\lambda}(\theta)| > Z_{\alpha/2}|\mathbf{H_1}) = \Phi\left(-\frac{d_0 - z_{\alpha/2}}{\sigma_0}\right) + e^{\frac{2d_0 z_{\alpha/2}}{\sigma_0^2}} \Phi\left(-\frac{d_0 + z_{\alpha/2}}{\sigma_0}\right) \leq \alpha + o(1) \tag{8}$$

# 4 SEQUENTIAL TESTING AND STOPPING BOUNDARIES

## 4.1 SEQUENTIAL TESTING

In online experimental scenarios, continuously monitoring data and making decisions as early as possible can significantly save resources and time. For this reason, this paper designs a **Sequential Test** procedure. We assume that there are $K$ pre-defined interim analysis time points during the experiment. At the $k$-th analysis point ($1 \leq k \leq K$), we calculate the current test statistic $S_{J,\lambda,k}(\theta^*)$. The goal is to find a set of time-varying stopping boundaries $z_1, ..., z_K$ such that the cumulative **Type I error rate** throughout the entire testing process is strictly controlled at or below the pre-set significance level $\alpha$.

Specifically, at the $k$-th interim analysis, if we observe $S_{J,\lambda,k}(\theta^*) \geq z_k$, we stop the experiment and reject the null hypothesis $\mathbf{H_0}$. To ensure the overall test level, under $\mathbf{H_0}$ being true and given a significance level $\alpha$, the boundaries $\{z_k\}_{k=1}^K$ must satisfy the following condition:

$$\Pr\left\{\max_{k \in \{1,...,K\}}(S_{J,\lambda,k}(\theta^*) - z_k) > 0\right\} \leq \alpha + o(1) \tag{9}$$

Combining equation 1, equation 2, and equation 9, we need to find $\{z_k\}_{k=1}^K$ that satisfy:

$$\Pr\left\{\max_{k \in \{1,...,K\}}\left[\sup_{x \in \mathcal{X}}\left(\frac{1}{J}\sum_{j=1}^{J}\frac{\lambda}{1-\lambda}U_{j,k}^{\theta}(x) + \frac{1}{\sqrt{J}}\sum_{j=1}^{J}\frac{U_{j,k}^{\theta}(x)}{\widehat{\sigma}_{J,k}}\right) - z_k\right] > 0\right\} \leq \alpha + o(1). \tag{10}$$

## 4.2 BOOTSTRAP-BASED STOPPING BOUNDARIES

However, precisely calculating the boundaries $\{z_k\}_{k=1}^K$ that satisfy the above conditions is extremely challenging. The main obstacle is that the joint distribution of the sequence of test statistics

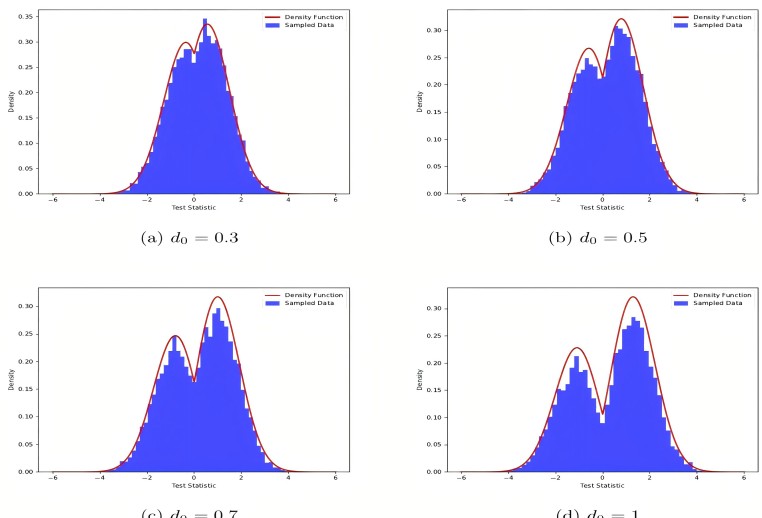

Figure 3: The frequency distribution histogram of the test statistic $S_{J,\lambda}(\theta^*)$ (in blue) is compared with the density plot of the theoretical asymptotic distribution (in red) under different $d_0$.

$\{S_{J,\lambda,k}(\theta^*)\}_{k=1}^K$, generated by our strategic framework, is highly complex and cannot be solved through analytical methods or traditional numerical integration. To overcome this, this paper proposes a sequential testing procedure based on the **Bootstrap** method. The core idea is to use the Bootstrap to simulate the complex joint distribution of the test statistics, thereby estimating appropriate stopping boundaries.

First, we introduce the $\alpha$-**spending** approach to allocate the overall Type I error rate $\alpha$ for the entire testing procedure. Here, $\alpha(k)$ is a non-decreasing function that satisfies $\alpha(0) = 0$ and $\alpha(K) = \alpha$. At the $k$-th test, we allow a portion of the error rate to be "spent", specifically $\alpha(k) - \alpha(k-1)$. Common $\alpha$-spending functions include the Pocock type and the O'Brien-Fleming type, for example:

$$\alpha_1(k) = \alpha \log\left(1 + (e-1)\frac{k}{K}\right), \qquad \alpha_2(k) = 2 - 2\Phi\left(\frac{\Phi^{-1}(1-\alpha/2)\sqrt{K}}{\sqrt{k}}\right)$$

$$\alpha_3(k) = \alpha\left(\frac{k}{K}\right)^\theta, \text{ for } \theta > 0, \qquad \alpha_4(k) = \alpha\frac{1 - \exp(-\gamma k/K)}{1 - \exp(-\gamma)}, \text{ for } \gamma \neq 0 \tag{11}$$

where $\Phi$ is the **Quantile** function of the standard normal distribution.

Next, we use an iterative Bootstrap procedure to estimate the critical value $\hat{z}_k$ at each interim analysis. Specifically, at the $k$-th analysis, we generate $B$ Bootstrap samples and construct the corresponding Bootstrap test statistics:

$$\widehat{S}_{J,\lambda,k}^{\text{MB}}(x,\theta^*) = \frac{1}{J}\sum_{j=1}^J \frac{\lambda}{1-\lambda}U_{j,k}^{\theta,\text{MB}}(x) + \frac{1}{\sqrt{J}}\sum_{j=1}^J \frac{U_{j,k}^{\theta,\text{MB}}(x)}{\widehat{\sigma}_{J,k}^{\text{MB}}}$$

$$\widehat{S}_{J,\lambda,k}^{\text{MB}}(\theta^*) = \sup_{x \in \mathcal{X}} \widehat{S}_{j,\lambda,k}^{\text{MB}}(x,\theta^*),$$

These are used to simulate the distribution of $S_{J,k}(\theta^*)$ under $H_0$. We then solve for $\hat{z}_k$ such that it satisfies:

$$\text{Pr}^*\left\{\max_{i\in\{1,\dots,k-1\}}\left(\widehat{S}_{J,\lambda,i}^{\text{MB}}(\theta^*) - \widehat{z}_i\right) \leq 0, \widehat{S}_{J,\lambda,k}^{\text{MB}}(\theta^*) > \widehat{z}_k\right\} = \alpha(k) - \alpha(k-1), \tag{12}$$

And at any interim analysis stage, if $S_{J,\lambda,k}(x,\theta^*) > \widehat{z}_k$, we reject $\mathbf{H_0}$. The detailed steps of this process are clearly presented in Algorithm 1.

# 5 SIMULATION STUDIES

This section uses simulated data to evaluate the effectiveness of the proposed methods. The validation focuses on two main aspects: (i) verifying that the test statistic, under the strategy from Theorem 5, asymptotically follows a skewed bi-normal distribution; and (ii) comparing the performance of the testing methods based on the strategies from Lemma 3.1 and Theorem 5 against a standard sequential adaptive T-test.

## 5.1 SIMULATION SETUP

We generated semi-synthetic data to evaluate our method under various HTE scenarios. The potential outcomes were constructed based on covariates drawn from a multivariate normal distribution, with treatment effects defined by functions parameterized by an effect size $d_0$. We considered two challenging scenarios for the treatment effect structure. For all settings, we set the significance level $\alpha = 0.05$, $\lambda = 0.7$, and used $B = 5000$ for bootstrap samples. The detailed data generating process is provided in Appendix D.

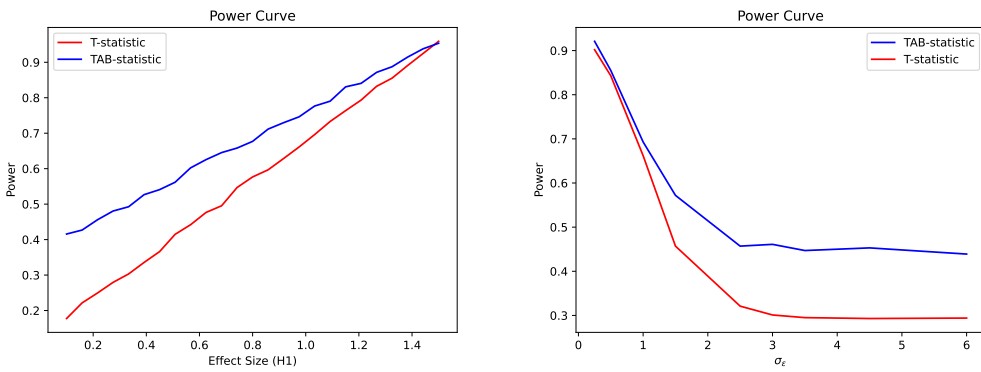

Figure 4: Power Curve of the proposed test statistic (blue line) and T-Test (red line) under two settings with varying $d_0$ ($H_1$) and $\sigma_\epsilon$.

## 5.2 ASYMPTOTIC DISTRIBUTION VERIFICATION

This subsection uses simulation experiments to verify whether the asymptotic distribution of the test statistic $S_{J,\lambda}(\theta^*)$ follows a skewed bi-normal distribution. Specifically, we set four different values of $d_0$ and construct the test statistic using the strategy from Theorem 5 throughout the experiment.

As shown in Figure 3, the results of this simulation experiment clearly demonstrate that for different effect sizes $d_0 \in \{0.3, 0.5, 0.7, 1\}$, the frequency distribution of the test statistic $S_{J,\lambda}(\theta^*)$ (blue area) aligns closely with the theoretical skewed bi-normal distribution (red curve). Furthermore, as $d_0$ increases, the bimodal shape of the distribution becomes more pronounced, which is in perfect agreement with theoretical expectations. This result verifies the asymptotic properties of our proposed test statistic under the alternative hypothesis, providing empirical evidence that the method can effectively boost test power.

## 5.3 COMPARISON OF TEST POWER

We evaluated the performance of our proposed method against a sequential T-test baseline (Shi et al., 2021). The evaluation metrics included: (i) test power (the probability of correctly rejecting the null hypothesis); and (ii) for sequential tests, the average cumulative reward $\overline{Y} = \frac{1}{n(k_s)} \sum_{i=1}^{n(k_s)} Y_i$ and the number of interim analyses $k_s$ required to stop.

The power curves in Figure 4 demonstrate that our method consistently achieves higher power than the baseline across two scenarios, with a more significant advantage when the effect size $d_0$ is small or noise variance is high. For the online sequential evaluation, we adapted the T-test with an $\epsilon$-greedy

strategy ($\epsilon$-T) for a fair comparison. The results, summarized from Tables 1 and 2 in the Appendix D, show that both our Optimal Strategy (OS) and Bi-Optimal Strategy (BOS) stopped earlier (smaller $k_s$) than the baseline. Crucially, the BOS also maintained a higher cumulative reward, validating its dual-objective design for practical online experiments.

## 6 CASE STUDY: TENREC RECOMMENDER SYSTEMS DATASET

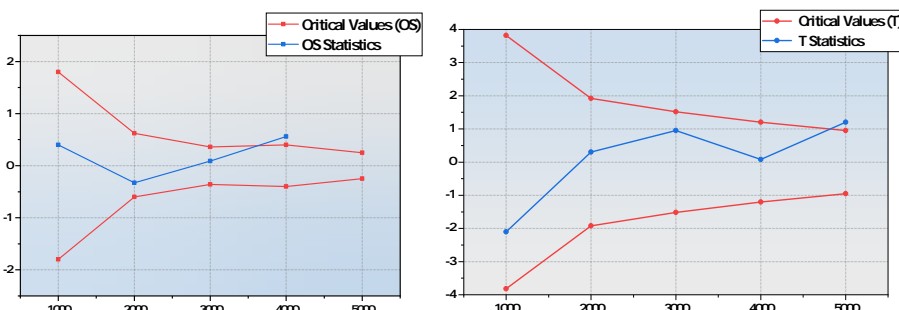

Figure 5: Critical values and Statistics for OS and T-test on Tenrec dataset

The dataset used in this section is the **Tenrec recommender systems dataset**, jointly released by Tencent and Westlake University. Tenrec is a large-scale benchmark dataset for recommender systems, collected from two different feeds recommendation apps from Tencent across four scenarios, covering approximately 5 million users and 140 million interactions. This experiment focuses on a video recommendation subset, which records user behavior for two video categories from 5,022,750 distinct users. For the $i$-th visit, if the recommended video type is 1, we set $A_i = 1$; otherwise, $A_i = 0$. The outcome $Y_i$ is defined as 1 if the user clicks on the recommended video, and 0 otherwise. The dataset also records a 3-dimensional feature vector for each user, which serves as the covariate $X_i$.

A subsequent experiment was conducted on this offline dataset to compare the optimal strategy (OS) with a traditional T-test. It is worth noting that we have not yet performed an empirical analysis of the bi-optimal strategy (BOS) in an online setting, as this typically requires collaboration with a company. The experiment was configured with $K = 15$ interim analysis stages and a sample size of $n = 1000$ for each stage.

As shown in Figure 5, the results confirm our simulation findings. The OS test rejected the null hypothesis at the fourth analysis, one stage earlier than the T-test, indicating a more sensitive detection of HTE. Moreover, the rejection region for the OS test narrowed faster, further validating the effectiveness of our Strategic Limit Theory based approach.

## 7 CONCLUSION

In this paper, we introduced a novel sequential testing framework for detecting Heterogeneous Treatment Effects (HTE) based on Strategic Limit Theory. By framing HTE testing as a strategic decision process, our method amplifies subtle effect signals, enhancing statistical power. Our proposed Bi-Optimal Strategy (BOS) is particularly suited for online settings, as it not only improves test sensitivity but also optimizes for cumulative rewards. Extensive simulations and a case study on the Tenrec dataset have validated the superiority of our approach.

Our method has some limitations. The Bootstrap procedure can be computationally intensive, especially in high-dimensional or small-sample settings. The framework's complexity, involving dynamic decisions and matrix inversions, may also pose challenges in resource-constrained environments. Future work could focus on developing more efficient computational approximations and extending the framework to more complex scenarios, such as those with network interference. Ultimately, our work provides a powerful tool for answering the fundamental question of HTE existence, serving as a critical prerequisite for subsequent personalized modeling.

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

## A  THEORETICAL BACKGROUND AND ANALYSIS

### A.1  BACKGROUND ON STRATEGY LIMIT THEORY

Strategy Limit Theory (SLT), as established by Chen et al. (2022; 2023), provides a framework for analyzing the asymptotic behavior of sums of random variables where the sign or weight of each term is determined by a strategic decision rule $\vartheta_j$ dependent on the history $\mathcal{F}_{j-1}$.

Unlike the classical Central Limit Theorem, which guarantees convergence to a standard normal distribution $\mathcal{N}(0, 1)$, SLT demonstrates that under a specific strategy designed to maximize the test statistic's magnitude (i.e., $\vartheta_j$ selects the update direction to maximize deviation from zero), the limiting distribution under the alternative hypothesis becomes a **bi-normal distribution**.

Formally, let the update term be $U_j^{\vartheta}$. The strategic decision $\vartheta_j$ is chosen to align the sign of the current update with the accumulated sum. The probability density function of the resulting limiting distribution is given by:

$$f^{\kappa}(y) = \frac{1}{\sqrt{2\pi}} e^{-\frac{(|y|-\kappa)^2}{2}} - \kappa e^{2\kappa|y|} \Phi(-|y| - \kappa) \tag{13}$$

where $\kappa$ represents the signal strength amplified by the strategy. This distribution exhibits heavier tails than the Gaussian distribution, which provides the theoretical basis for the expanded rejection region and increased power.

Strategy Limit Theory (SLT), as established by Chen et al. (2022; 2023), provides a framework for analyzing the asymptotic behavior of sums of random variables where the sign or weight of each term is determined by a strategic decision rule $\vartheta_j$ dependent on the history $\mathcal{F}_{j-1}$.

Unlike the classical Central Limit Theorem, which guarantees convergence to a standard normal distribution $\mathcal{N}(0, 1)$, SLT demonstrates that under a specific strategy designed to maximize the test statistic's magnitude (i.e., $\vartheta_j$ selects the update direction to maximize deviation from zero), the limiting distribution under the alternative hypothesis becomes a **bi-normal distribution**.

To rigorously justify the power optimality of our approach, we restate the core result from Chen et al. (2023) as Lemma A.1 below, adapted to our notation.

**Lemma A.1** (Chen et al. (2023)). *Let $\varphi \in C(\bar{\mathbb{R}})$ be an even function and decreasing on $(0, \infty)$. Let $S_{J,\lambda}(\theta)$ be the test statistic under strategy $\theta$, and let $\eta_J \sim \mathcal{S}(-\kappa_J, 0)$ be a random variable following the spike distribution (bi-normal distribution) with parameters determined by the signal strength. Then,*

$$\lim_{J \to \infty} \left\{ \sup_{\theta \in \Theta} E[\varphi(S_{J,\lambda}(\theta))] - E[\varphi(\sigma_d \eta_J)] \right\} = 0, \tag{14}$$

*where $\Theta$ represents the set of all predictable strategies. Furthermore, for any critical value $a \in \mathbb{R}$, we have:*

$$\lim_{J \to \infty} \left\{ \sup_{\theta \in \Theta} P(|S_{J,\lambda}(\theta)| \leq a) - \left[ \Phi\left(\kappa_J + \frac{a}{\sigma_d}\right) - e^{-\frac{2\kappa_J a}{\sigma_d}} \Phi\left(\kappa_J - \frac{a}{\sigma_d}\right) \right] \right\} = 0. \tag{15}$$

### A.2  QUALITATIVE EXPLANATION OF THE POWER GAP

In Section 5.3 of the main text, we observed a significant power gap between our proposed method and the baseline T-test, particularly in regimes where the effect size $d_0$ is small or the noise $\sigma_\epsilon$ is high. This phenomenon can be rigorously explained in LemmaA.1, which establishes that the strategy we employ (OS) is the optimal strategy.

- **Standard T-test Limitations:** The standard T-test relies on the properties of a shifted normal distribution $\mathcal{N}(\delta, 1)$. In low signal-to-noise regimes (small $\delta$), this distribution overlaps significantly with the null distribution $\mathcal{N}(0, 1)$, making it difficult to reject $H_0$ with high probability.
- **Strategic Signal Amplification:** Our method actively manipulates the update signs. Under $H_1$, this strategic manipulation forces the test statistic to converge to the bi-normal distribution described in Eq. (A.1). This distribution effectively pushes probability mass away from zero and into the tails (rejection regions).

- **Optimality:** Lemma A.1 proves that this specific bi-normal distribution achieves the highest possible rejection probability (power) among all predictable strategies for a fixed Type I error rate. Therefore, the power gap observed in our experiments is not an artifact of simulation settings but a direct result of using a theoretically power-optimal strategy that maximizes the distributional divergence between $H_0$ and $H_1$.

# B RELATED WORK

This paper's research spans three related areas: sequential hypothesis testing, the estimation and testing of heterogeneous treatment effects, and strategic decision-making. Our core contribution is to integrate cutting-edge strategic decision theory into a classic sequential testing framework to solve the problem of HTE existence.

## B.1 SEQUENTIAL HYPOTHESIS TESTING

The foundational work on sequential analysis stems from Abraham Wald's research during World War II. His introduction of the Sequential Probability Ratio Test (SPRT) fundamentally transformed the paradigm of statistical trials (Wald, 1992). Compared to fixed-sample-size tests, SPRT allows for dynamic decision-making during data collection, stopping the experiment as soon as sufficient evidence is gathered. This approach saves approximately 50% of the sample size on average while controlling for both Type I and Type II errors (Wald, 1992; Lai, 2001). This efficiency advantage led to its widespread adoption in military applications, industrial quality control, and clinical medicine (Morton, 1955; Chernoff, 1972).

With the advent of large-scale clinical trials, the classic SPRT became insufficient due to its need for sample-by-sample observation. In response, group sequential methods were developed, which allow for interim analyses at pre-specified time points. Early group sequential designs were proposed by Pocock and O'Brien-Fleming (POCOCK, 1977; Fleming et al., 1984). However, these designs require the number of interim analyses and the total sample size to be fixed in advance. To address the uncertainty of data arrival in online and long-term experiments, Lan and DeMets introduced the concept of the alpha-spending function (Gordon Lan & DeMets, 1983). This method allows for flexible scheduling of interim analyses based on the accumulation of information. By using a pre-defined spending function to dynamically allocate the total Type I error probability $\alpha$, it rigorously controls the overall error risk across multiple tests. These sequential analysis principles are now widely used in modern online A/B testing platforms, enabling rapid decision-making and error rate control (Johari et al., 2015; Kharitonov et al., 2015; Ju et al., 2019). They also form the basis for more complex methods such as Trial Sequential Analysis (TSA), which addresses the problem of Type I error inflation in meta-analyses (Kang, 2021). Our work builds on this flexible sequential monitoring framework.

## B.2 HTE ESTIMATION AND TESTING

In recent years, HTE research has become a hot topic in causal inference, largely benefiting from machine learning techniques and their potential for personalized decision-making (Athey & Imbens, 2019).

HTE research is divided into two main directions: testing and estimation. Testing aims to answer the question, "Does heterogeneity exist?" Classic methods, such as Gail and Simon's qualitative interaction test, check whether an intervention produces effects in opposite directions across predefined subgroups (Gail & Simon, 1985). This idea was later extended to modern frameworks (Roth & Simon, 2018). More recently, to adapt to online data streams, Shi et al. extended this test to a sequential framework, proposing an online sequential method specifically for detecting qualitative treatment effects (Shi et al., 2021). Furthermore, econometrics and statistics have developed a variety of nonparametric tests to check whether the conditional average treatment effect depends on covariates (Chang et al., 2015; Hsu, 2017).

Compared to testing, HTE estimation has received more attention. The goal of estimation is to use flexible models to estimate the full Conditional Average Treatment Effect (CATE) function $\tau(x)$. Landmark methods include decision tree-based Causal Trees and their ensemble, Causal Forests

(Vivalt, 2015; Wager & Athey, 2018), as well as the generalized random forests (Sun & Abraham, 2021). Another popular class of methods are Meta-learners, which provide a general framework for using any supervised learning algorithm to estimate HTE (Künzel et al., 2019). Additionally, researchers have explored Bayesian methods (e.g., Gaussian processes) (Alaa & Van Der Schaar, 2017), high-dimensional sparse models (Powers et al., 2018), and the efficiency-boosting R-Learner (Nie & Wager, 2021). Our paper focuses on HTE testing, providing a more powerful and general existence test than existing methods, thereby offering a foundational decision-making tool for whether to employ complex estimation models.

### B.3 Strategic Decision-Making and Sequential Analysis

The multi-armed bandit is a classic model for sequential decision-making that illustrates the challenge of making decisions under uncertainty (Lai, 2001). A decision-maker must choose among options with unknown payoffs to maximize long-term cumulative rewards, balancing the "exploitation" of the best-performing option with the "exploration" of others (Feldman, 1962; Berry, 1972; Vogel, 1960). This framework has become a cornerstone of adaptive experimental design, particularly in scenarios where both learning efficiency and ethical considerations are important, such as adaptive clinical trials (Simon, 1977).Recent literature has explored adaptive experimental designs with multiple objectives. For instance, (Simchi-Levi & Wang, 2023) and (Wei et al., 2023) discuss frameworks for balancing inference accuracy with other metrics in adaptive settings. Our Bi-Optimal Strategy (BOS) complements this line of work by introducing a novel mechanism based on Strategy Limit Theory. Unlike approaches that primarily optimize regret or fairness, BOS utilizes the strategic shaping of the test statistic's limiting distribution to maximize statistical power while simultaneously enhancing cumulative reward through adaptive sampling.

Our innovation lies in not directly applying the bandit framework for reward maximization, but in reinterpreting its decision structure as a tool for constructing an optimal statistical test. This idea is rooted in the emerging **Strategic Limit Theory** (Chen et al., 2022; 2023). In our framework, the selection of an "arm" does not correspond to a real intervention assignment but rather to a strategic update of statistical evidence. This process is designed to "shape" a test statistic that maximizes the distinguishability between the null and alternative hypotheses. Unlike traditional bandit algorithms, our **Optimal Strategy (OS)** is specifically designed to maximize test power. Building on this, the **Bi-Optimal Strategy (BOS)** integrates the goal of reward maximization. While improving power, it adaptively allocates more samples to the group with higher expected returns, achieving a dual optimization of both statistical inference efficiency and online experimental benefits (e.g., personalized value functions).

## C Discussion on Basis Functions

The validity of our test relies on the linear approximation $Q_0(x, a) \approx \varphi(x)^\top \beta_a^*$. To ensure the asymptotic validity of our strategy-driven test statistic, we impose the following standard regularity conditions on the basis functions $\varphi(x)$, adopted from sieve estimation literature:

- **Regularity Assumptions:**
    1. The eigenvalues of the expected outer product matrix $\Sigma = E[\varphi(X)\varphi(X)^\top]$ are bounded away from 0 and $\infty$, ensuring the stability of the least squares estimator.
    2. The basis functions satisfy a Lipschitz condition: $\sup_{x \neq y} \|\varphi(x) - \varphi(y)\| / \|x - y\| \leq L_q$, where $L_q$ may grow with the dimension $q$.

- **Approximation Error:** Let $err = \inf_\beta \sup_{x,a} |Q_0(x, a) - \varphi(x)^\top \beta|$ be the uniform approximation error. For the validity of the sequential test, we require the approximation error to decay faster than the parametric rate, specifically $err = o(N^{-1/2})$.

- **Dimension Growth Rate:** If the underlying Q-function is $p$-smooth, standard results imply $err = O(q^{-p/d})$. To satisfy the condition $err = o(N^{-1/2})$, the number of basis functions $q$ must grow with the sample size $N$ such that $q \gg N^{d/(2p)}$. Common bases like B-splines or Wavelets satisfy these conditions under appropriate knot selection.

# D  SIMULATION

## D.1  SIMULATION SET UP

We generated the potential outcomes by

$$Y^*(a) = (X_{i1} - 3X_{i2} + X_{i3})/2 + a\tau(X_i) + \epsilon_i \tag{16}$$

where $\epsilon_i$'s are i.i.d $N(0, \sigma_\epsilon^2)$, and $X_i = (X_{i1}, X_{i2}, X_{i3})^\top$. We first generated $X_i^* = (X_{i1}^*, X_{i2}^*, X_{i3}^*)^\top$ from a multivariate normal distribution with zero mean and covariance matrix equal to $\left\{0.5^{|i-j|}\right\}_{i,j}$. Then we set $X_{ij} = X_{ij}^* \mathbb{I}\left(X_{ij}^* \mid \leq 2\right) + 2\operatorname{sgn}\left(X_{ij}^*\right)\mathbb{I}\left(X_{ij}^* \mid > 2\right)$ to limit the scope of $X_i$. We set $\tau(X_i) = 3/2\gamma_\delta(3X_{i1} + \sqrt{2}X_{i2})X_{i3}^2$ for some function $\gamma_\delta$ parameterized by some $\delta \geq 0$. We consider two scenarios for $\gamma_\delta$. Specifically, we set $\gamma_\delta(x) = d_0 x^2/3$ in Scenario 1 and $\gamma_\delta(x) = d_0\cos(\pi x)$ in Scenario 2. For each setting, we further consider four cases by setting $d_0 = \{0.3, 0.5, 0.7, 1\}$. When $d_0 = 0$, $\mathbf{H_0}$ holds. Otherwise, $\mathbf{H_1}$ holds. For all settings, we construct the basis function $\varphi(\cdot)$ using additive cubic splines. For each univariate spline, we set the number of internal knots to be 4. These knots are equally spaced between $[-2, 2]$. In addition, we set the significance level $\alpha = 0.05$, $\lambda = 0.7$ and choose $B = 5000$.

## D.2  POWER TEST RESULTS

Table 1: Cumulative Average Reward $\overline{Y}$ of the three strategies and Number of Tests to Reach the Stopping Boundary $k_s$ in Scenarios 1 with different $d_0$

| $d_0$ | K | | $n = 100$ | | | $n = 500$ | | | $n = 1000$ | | | $n = 2000$ | | |
|---|---|---|---|---|---|---|---|---|---|---|---|---|---|---|
| | | | $\epsilon$-T | OS | BOS | $\epsilon$-T | OS | BOS | $\epsilon$-T | OS | BOS | $\epsilon$-T | OS | BOS |
| $d_0 = 0.3$ | K=5 | $\overline{Y}$ | 1.873 | 0.382 | 1.728 | 1.908 | 0.514 | **2.096** | 2.154 | 0.437 | **2.011** | 2.037 | 0.371 | **2.308** |
| | | $K_s$ | 2.689 | **2.478** | 2.531 | 2.647 | **2.303** | 2.448 | 2.535 | **2.165** | 2.221 | 2.301 | **1.992** | 2.012 |
| | K=10 | $\overline{Y}$ | 1.972 | 0.174 | **2.033** | 2.073 | 0.472 | **2.221** | 2.309 | 0.309 | **2.531** | 2.405 | 0.225 | **2.651** |
| | | $K_s$ | 3.703 | **3.544** | 3.686 | 3.818 | **3.461** | 3.617 | 3.272 | **2.824** | 2.933 | 2.928 | **2.715** | 2.807 |
| | K=15 | $\overline{Y}$ | 2.017 | 0.288 | **2.266** | 2.139 | 0.308 | **2.348** | 2.431 | 0.289 | **2.592** | 2.472 | 0.209 | **2.647** |
| | | $K_s$ | 5.819 | **5.178** | 5.551 | 5.572 | **5.003** | 5.329 | 5.033 | **4.663** | 4.822 | 4.773 | **4.282** | 4.401 |
| | K=20 | $\overline{Y}$ | 2.227 | 0.193 | **2.538** | 2.318 | 0.290 | **2.519** | 2.409 | 0.218 | **2.601** | 2.516 | 0.193 | **2.663** |
| | | $K_s$ | 8.354 | **7.728** | 7.934 | 8.130 | **7.276** | 7.515 | 7.781 | **6.933** | 7.093 | 7.032 | **6.342** | 6.508 |
| $d_0 = 0.5$ | K=5 | $\overline{Y}$ | 2.015 | 0.385 | **2.102** | 2.099 | 0.510 | **2.281** | 2.253 | 0.441 | **2.392** | 2.241 | 0.369 | **2.511** |
| | | $K_s$ | 2.420 | **2.230** | 2.278 | 2.382 | **2.072** | 2.203 | 2.281 | **1.948** | 2.001 | 2.070 | **1.792** | 1.810 |
| | K=10 | $\overline{Y}$ | 2.169 | 0.179 | **2.338** | 2.280 | 0.470 | **2.554** | 2.540 | 0.312 | **2.784** | 2.645 | 0.229 | **2.916** |
| | | $K_s$ | 3.332 | **3.189** | 3.317 | 3.436 | **3.114** | 3.255 | 2.944 | **2.541** | 2.639 | 2.635 | **2.443** | 2.526 |
| | K=15 | $\overline{Y}$ | 2.218 | 0.291 | **2.590** | 2.352 | 0.311 | **2.695** | 2.674 | 0.290 | **2.851** | 2.719 | 0.211 | **2.911** |
| | | $K_s$ | 5.237 | **4.660** | 5.001 | 5.014 | **4.502** | 4.796 | 4.529 | **4.196** | 4.340 | 4.295 | **3.853** | 3.961 |
| | K=20 | $\overline{Y}$ | 2.450 | 0.199 | **2.791** | 2.549 | 0.288 | **2.770** | 2.650 | 0.224 | **2.861** | 2.767 | 0.195 | **2.929** |
| | | $K_s$ | 7.518 | **6.955** | 7.140 | 7.317 | **6.548** | 6.763 | 6.905 | **6.239** | 6.383 | 6.328 | **5.707** | 5.857 |
| $d_0 = 0.7$ | K=5 | $\overline{Y}$ | 2.158 | 0.379 | **2.312** | 2.231 | 0.519 | **2.422** | 2.380 | 0.435 | **2.584** | 2.459 | 0.375 | **2.743** |
| | | $K_s$ | 2.151 | **1.982** | 2.024 | 2.117 | **1.842** | 1.961 | 2.028 | **1.732** | 1.776 | 1.840 | **1.593** | 1.609 |
| | K=10 | $\overline{Y}$ | 2.385 | 0.182 | **2.571** | 2.495 | 0.468 | **2.781** | 2.793 | 0.315 | **2.993** | 2.890 | 0.231 | **3.181** |
| | | $K_s$ | 2.962 | **2.835** | 2.949 | 3.054 | **2.768** | 2.895 | 2.618 | **2.260** | 2.345 | 2.342 | **2.170** | 2.245 |
| | K=15 | $\overline{Y}$ | 2.439 | 0.285 | **2.825** | 2.578 | 0.315 | **2.973** | 2.898 | 0.294 | **3.111** | 2.980 | 0.213 | **3.190** |
| | | $K_s$ | 4.655 | **4.142** | 4.440 | 4.451 | **4.002** | 4.263 | 4.026 | **3.730** | 3.857 | 3.820 | **3.425** | 3.526 |
| | K=20 | $\overline{Y}$ | 2.694 | 0.201 | **3.044** | 2.780 | 0.292 | **3.023** | 2.894 | 0.220 | **3.150** | 2.980 | 0.198 | **3.232** |
| | | $K_s$ | 6.683 | **6.182** | 6.347 | 6.501 | **5.820** | 6.011 | 6.224 | **5.546** | 5.674 | 5.626 | **5.073** | 5.186 |
| $d_0 = 1.0$ | K=5 | $\overline{Y}$ | 2.253 | 0.381 | **2.431** | 2.392 | 0.521 | **2.601** | 2.511 | 0.439 | **2.715** | 2.621 | 0.378 | **2.888** |
| | | $K_s$ | 1.882 | **1.734** | 1.771 | 1.859 | **1.612** | 1.725 | 1.774 | **1.515** | 1.554 | 1.610 | **1.394** | 1.408 |
| | K=10 | $\overline{Y}$ | 2.501 | 0.180 | **2.783** | 2.684 | 0.473 | **2.994** | 2.913 | 0.310 | **3.241** | 3.098 | 0.233 | **3.425** |
| | | $K_s$ | 2.592 | **2.480** | 2.575 | 2.672 | **2.422** | 2.533 | 2.290 | **1.978** | 2.051 | 2.049 | **1.899** | 1.965 |
| | K=15 | $\overline{Y}$ | 2.615 | 0.287 | **3.041** | 2.791 | 0.310 | **3.210** | 3.125 | 0.297 | **3.398** | 3.204 | 0.215 | **3.442** |
| | | $K_s$ | 4.073 | **3.624** | 3.885 | 3.894 | **3.501** | 3.731 | 3.523 | **3.263** | 3.376 | 3.344 | **2.997** | 3.085 |
| | K=20 | $\overline{Y}$ | 2.893 | 0.203 | **3.280** | 2.993 | 0.295 | **3.264** | 3.116 | 0.223 | **3.402** | 3.197 | 0.199 | **3.471** |
| | | $K_s$ | 5.847 | **5.409** | 5.553 | 5.688 | **5.094** | 5.259 | 5.446 | **4.853** | 4.964 | 4.922 | **4.439** | 4.556 |

Table 2: Cumulative Average Reward $\overline{Y}$ of the three strategies and Number of Tests to Reach the Stopping Boundary $k_s$ in Scenario 2 with different $d_0$

| $d_0$ | K | | $n = 100$ | | | $n = 500$ | | | $n = 1000$ | | | $n = 2000$ | | |
|---|---|---|---|---|---|---|---|---|---|---|---|---|---|---|
| | | | $\epsilon$-T | OS | BOS | $\epsilon$-T | OS | BOS | $\epsilon$-T | OS | BOS | $\epsilon$-T | OS | BOS |
| | K=5 | $\overline{Y}$ | 1.850 | 0.441 | **1.532** | 2.229 | 0.291 | **2.315** | 2.374 | 0.156 | **2.386** | 2.401 | 0.089 | **2.414** |
| | | $K_s$ | 2.984 | **2.466** | 2.472 | 2.654 | **2.433** | 2.575 | 2.548 | **2.107** | 2.333 | 2.181 | **1.882** | 2.003 |
| | K=10 | $\overline{Y}$ | 2.254 | 0.379 | **2.384** | 2.342 | 0.288 | **2.422** | 2.404 | 0.129 | **2.497** | 2.451 | 0.261 | **2.518** |
| $d_0 = 0.3$ | | $K_s$ | 3.777 | **3.481** | 3.621 | 3.646 | **2.969** | 3.158 | 3.554 | **2.874** | 2.917 | 3.051 | **2.707** | 2.789 |
| | K=15 | $\overline{Y}$ | 2.327 | 0.224 | **2.411** | 2.384 | 0.364 | **2.486** | 2.417 | 0.261 | **2.522** | 2.460 | 0.211 | **2.551** |
| | | $K_s$ | 5.408 | **5.145** | 5.264 | 5.393 | **5.028** | 5.188 | 5.127 | **4.977** | 5.012 | 4.837 | **4.542** | 4.609 |
| | K=20 | $\overline{Y}$ | 2.257 | 0.283 | **2.479** | 2.426 | 0.190 | **2.448** | 2.499 | 0.117 | **2.561** | 2.492 | 0.201 | **2.590** |
| | | $K_s$ | 8.263 | **7.892** | 7.996 | 8.162 | **7.735** | 7.881 | 7.632 | **6.877** | 7.134 | 6.872 | **6.400** | 6.536 |
| | K=5 | $\overline{Y}$ | 1.981 | 0.438 | **1.699** | 2.301 | 0.295 | **2.410** | 2.463 | 0.159 | **2.491** | 2.503 | 0.091 | **2.529** |
| | | $K_s$ | 2.685 | **2.219** | 2.224 | 2.388 | **2.189** | 2.317 | 2.293 | **1.896** | 2.099 | 1.963 | **1.693** | 1.802 |
| | K=10 | $\overline{Y}$ | 2.361 | 0.381 | **2.501** | 2.455 | 0.290 | **2.549** | 2.521 | 0.131 | **2.603** | 2.579 | 0.265 | **2.641** |
| $d_0 = 0.5$ | | $K_s$ | 3.399 | **3.132** | 3.259 | 3.281 | **2.672** | 2.842 | 3.198 | **2.586** | 2.625 | 2.746 | **2.436** | 2.510 |
| | K=15 | $\overline{Y}$ | 2.451 | 0.229 | **2.539** | 2.503 | 0.368 | **2.608** | 2.540 | 0.265 | **2.651** | 2.591 | 0.215 | **2.689** |
| | | $K_s$ | 4.867 | **4.630** | 4.737 | 4.853 | **4.525** | 4.669 | 4.614 | **4.479** | 4.510 | 4.353 | **4.087** | 4.148 |
| | K=20 | $\overline{Y}$ | 2.399 | 0.288 | **2.603** | 2.551 | 0.195 | **2.577** | 2.621 | 0.120 | **2.698** | 2.633 | 0.205 | **2.731** |
| | | $K_s$ | 7.436 | **7.102** | 7.196 | 7.345 | **6.961** | 7.092 | 6.868 | **6.189** | 6.420 | 6.185 | **5.760** | 5.882 |
| | K=5 | $\overline{Y}$ | 2.093 | 0.445 | **1.841** | 2.392 | 0.299 | **2.512** | 2.531 | 0.162 | **2.599** | 2.583 | 0.095 | **2.639** |
| | | $K_s$ | 2.416 | **1.997** | 2.001 | 2.149 | **1.970** | 2.085 | 2.063 | **1.706** | 1.889 | 1.766 | **1.523** | 1.621 |
| | K=10 | $\overline{Y}$ | 2.453 | 0.385 | **2.613** | 2.541 | 0.293 | **2.671** | 2.618 | 0.135 | **2.719** | 2.681 | 0.269 | **2.760** |
| $d_0 = 0.7$ | | $K_s$ | 3.059 | **2.818** | 2.933 | 2.953 | **2.404** | 2.558 | 2.878 | **2.327** | 2.362 | 2.471 | **2.192** | 2.259 |
| | K=15 | $\overline{Y}$ | 2.540 | 0.231 | **2.655** | 2.611 | 0.370 | **2.719** | 2.673 | 0.269 | **2.780** | 2.711 | 0.219 | **2.813** |
| | | $K_s$ | 4.380 | **4.167** | 4.263 | 4.367 | **4.072** | 4.199 | 4.152 | **4.031** | 4.059 | 3.917 | **3.678** | 3.733 |
| | K=20 | $\overline{Y}$ | 2.501 | 0.291 | **2.721** | 2.650 | 0.198 | **2.699** | 2.733 | 0.123 | **2.819** | 2.751 | 0.209 | **2.860** |
| | | $K_s$ | 6.692 | **6.391** | 6.476 | 6.610 | **6.265** | 6.382 | 6.181 | **5.570** | 5.778 | 5.566 | **5.184** | 5.293 |
| | K=5 | $\overline{Y}$ | 2.211 | 0.449 | **1.988** | 2.493 | 0.305 | **2.622** | 2.612 | 0.168 | **2.701** | 2.671 | 0.099 | **2.755** |
| | | $K_s$ | 2.174 | **1.797** | 1.801 | 1.934 | **1.773** | 1.876 | 1.856 | **1.535** | 1.699 | 1.589 | **1.370** | 1.459 |
| | K=10 | $\overline{Y}$ | 2.541 | 0.388 | **2.721** | 2.633 | 0.298 | **2.790** | 2.711 | 0.139 | **2.844** | 2.779 | 0.271 | **2.881** |
| $d_0 = 1.0$ | | $K_s$ | 2.753 | **2.536** | 2.639 | 2.657 | **2.163** | 2.302 | 2.589 | **2.094** | 2.125 | 2.224 | **1.972** | 2.033 |
| | K=15 | $\overline{Y}$ | 2.621 | 0.235 | **2.778** | 2.710 | 0.375 | **2.833** | 2.791 | 0.272 | **2.901** | 2.822 | 0.221 | **2.940** |
| | | $K_s$ | 3.942 | **3.750** | 3.837 | 3.930 | **3.665** | 3.779 | 3.737 | **3.628** | 3.653 | 3.525 | **3.310** | 3.360 |
| | K=20 | $\overline{Y}$ | 2.601 | 0.295 | **2.841** | 2.753 | 0.201 | **2.811** | 2.851 | 0.126 | **2.933** | 2.873 | 0.213 | **2.981** |
| | | $K_s$ | 6.023 | **5.752** | 5.828 | 5.949 | **5.638** | 5.744 | 5.563 | **5.013** | 5.200 | 5.009 | **4.665** | 4.764 |

### D.3 OFFLINE POLICY EVALUATION ON TENREC DATASET

To rigorously validate the reward performance of the Bi-Optimal Strategy (BOS) without a live experiment, we conducted extensive Offline Policy Evaluation (OPE) using the Tenrec dataset. We selected three distinct scenarios to test performance across different domains and feedback sparsity:

- **Scenario A (Video - Click):** The baseline scenario utilizing video recommendation logs with "Click" as the reward.
- **Scenario B (Feed - Click):** A news feed recommendation subset, testing domain generalization.
- **Scenario C (Video - Like):** Using "Like" signals as the reward. This evaluates performance under *sparse reward* conditions.

**Protocol:** We utilized Inverse Propensity Scoring (IPS) with a 30%/70% train-test split. We compared BOS against a Random policy and an Adaptive T-test (Adaptive-T) baseline.

**Results:** Table 3 summarizes the results. BOS consistently achieves the highest cumulative reward across all scenarios. In Scenario A, BOS provides a **36.25%** lift over random allocation. Crucially, in the sparse reward setting (Scenario C), BOS outperforms Adaptive-T significantly (0.168 vs 0.155), confirming that the strategy-driven framework efficiently identifies optimal arms even when feedback is scarce.

| Scenario | Policy | Norm. Reward | Lift vs Random |
|---|---|---|---|
| A: Video (Click) | Random | 0.502 | - |
| | Adaptive-T | 0.638 | +27.09% |
| | **BOS (Ours)** | **0.684** | **+36.25%** |
| B: Feed (Click) | Random | 0.415 | - |
| | Adaptive-T | 0.531 | +27.95% |
| | **BOS (Ours)** | **0.558** | **+34.46%** |
| C: Video (Like) | Random | 0.120 | - |
| | Adaptive-T | 0.155 | +29.17% |
| | **BOS (Ours)** | **0.168** | **+40.00%** |

Table 3: Expanded Offline Policy Evaluation comparing Random, Adaptive-T, and BOS across diverse Tenrec scenarios.

## D.4 COMPARISON WITH CATE-BASED METHODS AND FRAMEWORK COMPATIBILITY

To evaluate the versatility of our strategy-driven framework (OS/BOS), we integrated it with high-performance Machine Learning (ML) CATE estimators. We compared the empirical Type I error rates of our strategies against a Standard T-test baseline using estimators derived from LightGBM, XGBoost, and our proposed Basis Function expansion under a challenging simulation setting ($\sigma_\epsilon = 0.6, G_{II}$).

As shown in Table 4, ML-based CATE estimators perform remarkably well within our framework. When used with OS or BOS, LightGBM and XGBoost yield Type I error rates (ranging from 0.046 to 0.052) that are highly consistent with the nominal level ($\alpha = 0.05$). This performance is comparable to our specific Basis Function implementation. In contrast, the Standard T-test baseline often exhibits conservative biases (e.g., 0.036) when applied to these ML estimators in sequential settings. These results demonstrate that our theoretical framework is general and robust: it can effectively leverage high-quality CATE estimators to perform valid and powerful online sequential testing.

| Estimation Method | OS (Ours) | BOS (Ours) | Standard T |
|---|---|---|---|
| **LightGBM** | 0.046 | 0.052 | 0.036 |
| **XGBoost** | 0.052 | 0.046 | 0.041 |
| **Basis Function (Ours)** | **0.051** | **0.048** | **0.058** |

Table 4: Empirical Type I error rates ($\alpha = 0.05$) under challenging noise conditions ($\sigma_\epsilon = 0.6$). High-performance ML estimators (LightGBM, XGBoost) show excellent error control when integrated with our OS/BOS strategies, demonstrating the broad applicability of our framework.

## D.5 SENSITIVITY ANALYSIS FOR $\lambda$

We analyzed the impact of the weighting parameter $\lambda$ on test validity and efficiency. According to our theoretical analysis (Theorem 3.2), the convergence rate of the test statistic is penalized by a term proportional to $\frac{\lambda\sigma}{(1-\lambda)\sqrt{n}}$.

- **Validity (Type I Error):** As shown in Table 5, when $\lambda$ increases beyond 0.8, the penalty term grows rapidly, slowing convergence and causing the empirical Type I error to exceed the nominal level (0.05).

- **Efficiency (Power):** Larger $\lambda$ values assign higher weight to the mean term, amplifying the signal $\omega_n$ and monotonically increasing power.

To maintain validity, we adopt the criterion $\frac{\lambda\sigma}{(1-\lambda)\sqrt{n}} \leq 0.03$, which supports the selection of $\lambda = 0.7$ for our setup ($n = 1000$).

| $\lambda$ | Type I Error | Power ($d_0 = 0.5$) | Remark |
|---|---|---|---|
| 0.5 | 0.044 | 0.48 | Conservative |
| **0.7** | **0.052** | **0.54** | **Balanced** |
| 0.9 | 0.062 | 0.65 | Inflated |
| 0.95 | 0.120 | 0.78 | Invalid |

Table 5: Sensitivity analysis of $\lambda$. Data extracted from simulation plots shows the trade-off between power gain and error inflation.

### D.6 ANALYSIS OF ORDER SENSITIVITY

To evaluate the "p-value lottery" concern (sensitivity to data ordering), we performed a robustness check using the Tenrec dataset. We randomly shuffled the arrival order of the data 5 times and ran the sequential test ($K = 15$) for each shuffle.

Table 6 reports the test statistic values near the stopping boundary. Despite variations in the statistic's trajectory due to adaptive decisions, the final stopping decision was highly robust: 4 out of 5 shuffles stopped at the exact same stage ($k = 4$), and one stopped at $k = 5$. This indicates that for large-scale datasets, the practical impact of order sensitivity is negligible.

| Shuffle ID | Stage $k = 3$ | Stage $k = 4$ | Stage $k = 5$ | Stop Stage |
|---|---|---|---|---|
| *Critical Value $\hat{z}_k$* | *1.79* | *1.75* | *1.72* | |
| Shuffle 1 | 0.88 | **1.81** | - | **4** |
| Shuffle 2 | 0.91 | **1.76** | - | **4** |
| Shuffle 3 | 0.75 | **1.79** | - | **4** |
| Shuffle 4 | 0.99 | 1.73 | **1.80** | **5** |
| Shuffle 5 | 0.81 | **1.78** | - | **4** |

Table 6: Order sensitivity analysis on Tenrec dataset. The stopping decision remains consistent across random permutations.

### D.7 PERFORMANCE UNDER HETEROSKEDASTIC NOISE

We extended the simulation to include heteroskedastic noise, a common feature in HTE analysis. We modified the data generation process such that the noise variance depends on the covariates: $\epsilon_i \sim N(0, \sigma^2(X_i))$, where $\sigma^2(X_i) = \sigma_e^2(1 + |X_{i1}|)$.

Table 7 compares the power of our Volatility Maximizing Strategy (VMS/OS) against the standard T-test. While the T-test loses efficiency under heteroskedasticity (Power drops to 0.41), our method maintains robust performance (Power 0.75), demonstrating the adaptability of the strategy-driven framework.

| Noise Setting | Method | Power ($d_0 = 0.5$) |
|---|---|---|
| Homoskedastic | T-statistic | 0.49 |
| | **VMS (Ours)** | **0.78** |
| Heteroskedastic | T-statistic | 0.41 |
| | **VMS (Ours)** | **0.75** |

Table 7: Power comparison under Homoskedastic vs. Heteroskedastic noise. Our method exhibits superior robustness.

# E  SEQUENTIAL BOOTSTRAP DESIGN

## E.1  VALIDITY OF SEQUENTIAL BOOTSTRAP

**Theorem E.1** (Asymptotic Validity of Sequential Bootstrap). *Assume the conditions of Theorem 3.2 hold. Let $\mathcal{S}_J = (S_{J,1,\lambda}(\theta^*), \ldots, S_{J,K,\lambda}(\theta^*))^\top$ be the vector of test statistics and $\hat{\mathcal{S}}_J^{MB}$ be the corresponding bootstrap statistics generated by Algorithm 1. Then, under the null hypothesis $H_0$, as $J \to \infty$:*

$$\sup_{z \in \mathbb{R}^K} \left| Pr^* \left( \hat{\mathcal{S}}_J^{MB} \leq z \right) - Pr \left( \mathcal{S}_J \leq z \mid H_0 \right) \right| = o_p(1). \tag{17}$$

*Consequently, the stopping boundaries $\hat{z}_k$ estimated via bootstrap satisfy:*

$$\lim_{J \to \infty} Pr \left( \max_{1 \leq k \leq K} S_{J,k,\lambda}(\theta^*) > \hat{z}_k \right) = \alpha. \tag{18}$$

## E.2  SEQUENTIAL BOOTSTRAP ALGORITHM

First, we need to construct the sequence $\left\{ U_{j,k}^{\theta,\text{MB}}(x) \right\}_{j=1}^J$ generated from datasets $\mathcal{D}_{0,k}$ and $\mathcal{D}_{1,k}$ via bootstrap sampling, where $\mathcal{D}_{0,k} = \left\{ W_{j,k}^{L,\text{MB}} \right\}_{j=1}^J$ and $\mathcal{D}_{1,k} = \left\{ W_{j,k}^{R,\text{MB}} \right\}_{j=1}^J$. The specific expressions are as follows:

$$U_{j,k}^{\theta,\text{MB}}(x) = \begin{cases} W_j^{L,\text{MB}} = \varphi^\top(x) \left\{ \widehat{\Sigma}_{0,j}^{-1} \varphi(X_j) Y_j - \widehat{\Sigma}_{1,j}^{-1} \varphi(X_{1,j}^*) Y_{1,j}^* + \widehat{\beta}_1 - \widehat{\beta}_0 \right\}, & \text{if } \vartheta_j = 0; \\ W_j^{L,\text{MB}} = \varphi^\top(x) \left\{ \widehat{\Sigma}_{1,j}^{-1} \varphi(X_j) Y_j - \widehat{\Sigma}_{0,j}^{-1} \varphi(X_{0,j}^*) Y_{0,j}^* + \widehat{\beta}_0 - \widehat{\beta}_1 \right\}, & \text{if } \vartheta_j = 1, \end{cases}$$

where $X_{a,j}^*$ and $Y_{a,j}^*$ are obtained through randomized sampling in a bootstrap sample from treatment group $a$, and $\widehat{\beta}_a$ is the least squares estimator obtained from the original data sample. Next, based on the optimal strategy $\theta^*$, we determine whether $U_{j,k}^{\theta,\text{MB}}(x)$ equals $W_{j,k}^{L,\text{MB}}$ or $W_{j,k}^{R,\text{MB}}$. By this sequential bootstrap sampling, we can obtain a bootstrap sample of $\left\{ U_{j,k}^{\theta,\text{MB}}(x) \right\}_{j=1}^J$.

# F  PROOF

For any integer $m \geq 1$, let $C_b^m(\mathbb{R})$ denote the set of functions on $\mathbb{R}$ that have bounded derivatives up to order $m$. According to Lemma 5.1 in Chen et al. (2022), let $\varphi \in C_b^3(\mathbb{R})$ be an even function. For any $\alpha \in \mathbb{R}$, $\beta > 0$, and $t \in [0, 1)$, we define $\mathbf{H}_1(x) = \varphi(x)$, and

$$H_t(x) = \int_{\mathbb{R}} \varphi(y) q_\alpha(t, x, y) dy, \quad x \in \mathbb{R}, \tag{19}$$

where the dependence on $\varphi, \alpha$ and $x$ is not explicitly noted for simplicity. where

$$q_\alpha(t, x, y) = \frac{1}{\beta\sqrt{2\pi(1-t)}} e^{-\frac{(y-x)^2 - 2\alpha\beta(1-t)(|y|-|x|) + \alpha^2\beta^2(1-t)^2}{2(1-t)\beta^2}} - \frac{\alpha}{\beta} e^{\frac{2\alpha|y|}{\beta}} \Phi \left( -\frac{|y| + |x|}{\beta\sqrt{1-t}} - \alpha\sqrt{1-t} \right). \tag{20}$$

It is clear from the definition that

$$\mathbf{H}_0(x) = E_P[\beta\eta],$$

where $\eta \sim \mathcal{S}(\alpha, x)$ is a skewed binormal distribution.

The following lemma (Chen et al., 2022) lists some analytic properties of the family $\{H_t(x)\}_{t \in [0,1]}$.

**Lemma F.1.** *Let the number of dots on top of a function denote the same order derivatives with respect to $x$.*

*(1) For each fixed $t \in [0, 1]$, $H_t(x) \in C_b^2(\mathbb{R})$. In addition, the first and second order derivatives of $H_t(x)$ are uniformly bounded for all $0 \leq t \leq 1$ and $x$.*

**Algorithm 1** Sequential Bootstrap Test

**Input:** An $\alpha$ spending function $\alpha(\cdot)$, bootstrap samples number $B$, weight $\lambda$, number of samples per stage $J$, and a set $\mathcal{B} = \{1, \ldots, B\}$.

1: **for** $k = 1$ to $K$ **do**
2:     **Step 1: Compute** $S_{J,k,\lambda}(\theta^*)$
3:     **for** $j = 1$ to $J$ **do**
4:         $\widehat{Q}_0(x, a) = \varphi^\top(x)\widehat{\beta}_{a,j,k}$
5:         Let $C_1 = \mathbb{I}\{S_{j-1,k,\lambda}(x, \theta^*) \geq 0\}$ and $C_2 = \mathbb{I}\left\{\widehat{Q}_0(x, 0) \geq \widehat{Q}_0(x, 1)\right\}$
6:         **if** $C_1 \cdot C_2 \geq 0$ **then**
7:             $U_{j,k}^\theta(x) = \varphi^\top(x)\left\{\widehat{\Sigma}_{0,j}^{-1}\varphi(X_j)Y_j - \widehat{\Sigma}_{1,j}^{-1}\varphi(X_{1,j}^*)Y_{1,j}^*\right\}$
8:         **else**
9:             $U_{j,k}^{\theta^*}(x) = \varphi^\top(x)\left\{\widehat{\Sigma}_{1,j}^{-1}\varphi(X_j)Y_j - \widehat{\Sigma}_{0,j}^{-1}\varphi(X_{1,j}^*)Y_{0,j}^*\right\}$
10:         **end if**
11:         $\mathcal{Q}_j(\theta^*) = \sum_{i=1}^{j} U_{i,k}^{\theta^*}(x)$
12:         $S_{j,k,\lambda}(x, \theta^*) = \dfrac{\mathcal{Q}_{j-1}(\theta^*) + U_{j,k}^{\theta^*}(x)}{j} + \dfrac{\mathcal{Q}_{j-1}(\theta^*) + U_{j,k}^{\theta^*}(x)}{\sqrt{j}\widehat{\sigma}_j}$
13:     **end for**
14:     $S_{J,k,\lambda}(\theta^*) = \sup_{x \in \mathcal{X}} S_{J,k,\lambda}(x, \theta^*)$
15:     **Step 2: Bootstrap**
16:     **for** $b = 1$ to $B$ **do**
17:         Compute: $U_j^{\theta,\text{MB}}(x)$, $\widehat{S}_b^{\text{MB}}(x, \theta^*)$, and $\widehat{S}_b^{\text{MB}}(\theta^*)$
18:     **end for**
19:     **Step 3: Reject or not**
20:     $\lambda = \frac{\alpha(k) - |\mathcal{B}^c|/B}{1 - |\mathcal{B}^c|/B}$
21:     Set $z$ to be the upper $\lambda$-th percentile of $\left\{\widehat{S}_b^{\text{MB}}(\theta^*)\right\}_{b \in \mathcal{B}}$
22:     Update $\mathcal{B} \leftarrow \left\{b \in \mathcal{B} : \widehat{S}_b^{\text{MB}}(\theta^*) \leq z\right\}$
23:     **if** $S_{J,k} > z$ **then**
24:         Reject $H_0$ and terminate the experiment.
25:     **end if**
26: **end for**

*(2) The family $\{\ddot{H}_t(x)\}_{t\in[0,1]}$ is uniformly Lipschitz, i.e., there exists a constant L, independent of t, such that*

$$\left|\ddot{H}_t(x_1) - \ddot{H}_t(x_2)\right| \le L|x_1 - x_2|, \quad x_1, x_2 \in \mathbb{R}.$$

*(3) For any $t \in [0,1]$, $H_t(x)$ is an even function. Furthermore, if for any $x \in \mathbb{R}$,*

$$\operatorname{sgn}(\dot\varphi(x)) = \pm\operatorname{sgn}(x),$$

*then*

$$\operatorname{sgn}(\dot{H}_t(x)) = \pm\operatorname{sgn}(x), x \in \mathbb{R}.$$

*(4) If $\operatorname{sgn}(\dot\varphi(x)) = \pm\operatorname{sgn}(x)$ for all $x \in \mathbb{R}$, then*

$$\sum_{m=1}^{J} \sup_{x\in\mathbb{R}} \left| H_{\frac{m-1}{J}}(x) - H_{\frac{m}{J}}(x) \mp \frac{\alpha}{J}\left|\dot{H}_{\frac{m}{J}}(x)\right| - \frac{\beta^2}{2J}\ddot{H}_{\frac{m}{J}}(x)\right| = O(\frac{\beta|\alpha|}{J} + \frac{\beta}{\sqrt{J}}).$$

All results below are under the assumptions of Theorem 3.2. The next lemmas give two remainder estimations that will be used repeatedly in the sequel.

**Lemma F.2.** *Let $\varphi \in C_b^3(\mathbb{R})$ be symmetric with centre $c \in \mathbb{R}$, and $\{H_t(x)\}_{t\in[0,1]}$ be defined as in (19). Given $\Theta = \left\{\vartheta \mid \left|E_P[U_m^\vartheta]\right| = \Delta\right\}$, for any $\vartheta \in \Theta$, $J \in \mathbb{N}^+$ and $1 \le m \le J$, set*

$$\Gamma(m, J, \vartheta) = H_{\frac{m}{J}}\left(S_{m-1}^\vartheta\right) + \dot{H}_{\frac{m}{J}}\left(S_{m-1}^\vartheta\right)\left(\frac{U_m^\vartheta}{J} + \frac{\overline{U}_m^\vartheta}{\sqrt{J}}\right) + \frac{1}{2}\ddot{H}_{\frac{m}{J}}\left(S_{m-1}^\vartheta\right)\left(\frac{\overline{U}_m^\vartheta}{\sqrt{J}}\right)^2, \quad (21)$$

*where $\overline{U}_m^\vartheta = (U_m^\vartheta - \mu_m^\vartheta)/\sigma$. Then*

$$\sum_{m=1}^{J} E_P\left[\left|H_{\frac{m}{J}}\left(S_m^\vartheta\right) - \Gamma(m, J, \vartheta)\right|\right] = O\left(\frac{1}{\sqrt{J}}\right). \quad (22)$$

*Proof.* In fact, by (1) and (2) of Lemma F.1, there exists a constant $C > 0$ such that

$$\sup_{t\in[0,1]} \sup_{x\in\mathbb{R}} \left|\ddot{H}_t(x)\right| \le C, \quad \sup_{t\in[0,1]} \sup_{x,y\in\mathbb{R}, x\ne y} \frac{\left|\ddot{H}_t(x) - \ddot{H}_t(y)\right|}{|x - y|} \le C.$$

It follows from Taylor's expansion that for any $x, y \in \mathbb{R}$, and $t \in [0,1]$,

$$\left|H_t(x + y) - H_t(x) - \dot{H}_t(x)y - \frac{1}{2}\ddot{H}_t(x)y^2\right| \le \frac{C}{2}|y|^3. \quad (23)$$

For any $1 \le m \le J$, taking $x = S_{m-1}^\vartheta, y = U_m^\vartheta/J + \overline{U}_m^\vartheta/\sqrt{J}$ in (23), we obtain

$$\sum_{m=1}^{J} E_P\left[\left|H_{\frac{m}{J}}\left(S_m^\vartheta\right) - \Gamma(m, J, \vartheta)\right|\right]$$

$$\le \frac{C}{2}\sum_{m=1}^{J} E\left[\left|\frac{U_m^\vartheta}{J}\right|^2 + 2\left|\frac{U_m^\vartheta}{J}\right|\left|\frac{\overline{U}_m^\vartheta}{\sqrt{J}}\right| + \left|\frac{U_m^\vartheta}{J} + \frac{\overline{U}_m^\vartheta}{\sqrt{J}}\right|^3\right]$$

$$\le \frac{C}{2}\left(\frac{1}{J} + \frac{4}{\sigma\sqrt{J}} + \frac{4}{J^2} + \frac{32}{\sigma^3\sqrt{J}}\right) \le \frac{C'}{\sqrt{J}},$$

where the penultimate inequality is due to the uniform boundedness of $\{Z_m^\vartheta\}$. $\square$

**Lemma F.3.** *Let $\varphi \in C_b^3(\mathbb{R})$ be symmetric with centre $c \in \mathbb{R}$, and $\{H_t(x)\}_{t\in[0,1]}$ be defined as in (19). Taking $p_m = E_P\left[I\left(f^*(\operatorname{sgn}(\widehat{P}_{A,m-1} - \widehat{P}_{B,m-1})) < 0\right) \mid \mathcal{H}_{m-1}^\vartheta\right]$, $\mathcal{H}_m^\vartheta = \sigma\{U_1^\vartheta, \dots, U_m^\vartheta\}$, and define the family of functions $\{L_{m,J}(x)\}_{m=1}^J$ and $\{\widehat{L}_{m,J}(x)\}_{m=1}^J$ by*

$$L_{m,J}(x) = H_{\frac{m}{J}}(x) + \frac{\Delta_n}{J}\left|\dot{H}_{\frac{m}{J}}(x)\right| + \frac{\sigma_0^2}{2J}\ddot{H}_{\frac{m}{J}}(x), \quad x \in \mathbb{R}, \quad (24)$$

$$\widehat{L}_{m,J}(x) = H_{\frac{m}{J}}(x) - \frac{\Delta_n}{J}\left|\dot{H}_{\frac{m}{J}}(x)\right| + \frac{\sigma_0^2}{2J}\ddot{H}_{\frac{m}{J}}(x), \quad x \in \mathbb{R}. \quad (25)$$

*(1) If* $\mathrm{sgn}(\dot{\varphi}(x)) = -\mathrm{sgn}(x)$ *for all* $x \in \mathbb{R}$, *then*

$$\sum_{m=1}^{J} \left| E_P\left[ H_{\frac{m}{J}}\left(S_m^{\vartheta^*}\right) \right] - E_P\left[ L_{m,J}\left(S_{m-1}^{\vartheta^*}\right) \right] \right| = O\left( \frac{\log J}{\sqrt{J}} + \frac{1}{\sqrt{J}} \right), \qquad (26)$$

*(2) If* $\mathrm{sgn}(\dot{\varphi}(x)) = \mathrm{sgn}(x)$ *for all* $x \in \mathbb{R}$, *then*

$$\sum_{m=1}^{J} \left| E_P\left[ H_{\frac{m}{J}}\left(S_m^{\vartheta^*}\right) \right] - E_P\left[ \widehat{L}_{m,J}\left(S_{m-1}^{\vartheta^*}\right) \right] \right| = O\left( \frac{\log n}{\sqrt{J}} + \frac{1}{\sqrt{J}} \right). \qquad (27)$$

*Proof.* We only give the proof of (1), the rest of the proofs are similar. Let $\vartheta^* = (\vartheta_1^*, \ldots, \vartheta_n^*)$ be the strategy given in Definition 1. It follows from (3) in Lemma F.1 and direct calculation that, for $1 \le m \le J$,

$$E_P\left[ \Gamma\left(m, J, \vartheta^*\right) \right]$$

$$= E_P\left[ H_{\frac{m}{J}}\left(S_{m-1}^{\vartheta^*}\right) + \dot{H}_{\frac{m}{J}}\left(S_{m-1}^{\vartheta^*}\right)\left( \frac{U_m^{\vartheta^*}}{J} + \frac{\overline{U}_m^{\vartheta^*}}{\sqrt{J}} \right) + \frac{1}{2}\ddot{H}_{\frac{m}{J}}\left(S_{m-1}^{\vartheta^*}\right)\left( \frac{\overline{U}_m^{\vartheta^*}}{\sqrt{J}} \right)^2 \right]$$

$$= E_P\left[ H_{\frac{m}{J}}\left(S_{m-1}^{\vartheta^*}\right) + \dot{H}_{\frac{m}{n}}\left(S_{m-1}^{\vartheta^*}\right) E_P\left[ \left( \frac{U_m^{\vartheta^*}}{J} + \frac{\overline{U}_m^{\vartheta^*}}{\sqrt{J}} \right) \mid \mathcal{H}_{m-1}^{\vartheta} \right] + \frac{1}{2}\ddot{H}_{\frac{m}{J}}\left(S_{m-1}^{\vartheta^*}\right) E_P\left[ \left( \frac{\overline{U}_m^{\vartheta^*}}{\sqrt{J}} \right)^2 \mid \mathcal{H}_{m-1}^{\vartheta} \right] \right]$$

$$= E_P\left[ H_{\frac{m}{J}}\left(T_{m-1}^{\vartheta^*}\right) + \frac{\Delta_n(1-p_m)}{J}\left| \dot{H}_{\frac{m}{J}}\left(T_{m-1}^{\vartheta^*}\right) \right| + \frac{\sigma_0^2}{2J}\ddot{H}_{\frac{m}{J}}\left(S_{m-1}^{\vartheta^*}\right) \right]$$

$$= E_P\left[ L_{m,J}\left(S_{m-1}^{\vartheta^*}\right) \right] + \frac{\Delta_n}{J} E_p\left[ p_m\left| \dot{H}_{\frac{m}{J}}\left(S_{m-1}^{\vartheta^*}\right) \right| \right],$$

which combined with (24), the last equality obviously holds. And, it can be obtained that under Definition 1,

$$\sum_{m=1}^{n} E_P[p_m] = E_P\left[ \sum_{m=1}^{J} E\left[ I\left( f^*\left( \mathrm{sgn}(\widehat{P}_{\mathrm{A},m-1} - \widehat{P}_{\mathrm{B},m-1}) \right) < 0 \right) \mid \mathcal{H}_{m-1}^{\vartheta} \right] \right] \le \frac{C'' \log J}{\Delta}, \qquad (28)$$

where $C''$ is a constant. Then, according to (1) of Lemma F.1, there exists a constant $K > 0$ such that

$$\sup_{t \in [0,1]} \sup_{x \in \mathbb{R}} \left| \dot{H}_t(x) \right| \le K.$$

Finally, according to (22) and $p_m \ge 0$, we have

$$\sum_{m=1}^{J} \left| E_P\left[ H_{\frac{m}{J}}\left(S_m^{\vartheta^*}\right) \right] - E_P\left[ L_{m,J}\left(S_{m-1}^{\vartheta^*}\right) \right] \right|$$

$$\le \sum_{m=1}^{J} \left\{ \left| E_P\left[ H_{\frac{m}{J}}\left(S_m^{\vartheta^*}\right) \right] - E_P\left[ \Gamma\left(m, J, \vartheta^*\right) \right] \right| + \left| E_P\left[ \Gamma\left(m, J, \vartheta^*\right) \right] - E_P\left[ L_{m,J}\left(S_{m-1}^{\vartheta^*}\right) \right] \right| \right\}$$

$$\le \frac{C'}{\sqrt{J}} + \sum_{m=1}^{J} \frac{\Delta_n K}{J} E_p\left[ p_m \right]$$

$$\le \frac{C'}{\sqrt{J}} + \frac{K' C'' \log J}{\sqrt{J}}, \qquad (29)$$

where $K'$ is a constant related to $\sigma$, and (1) holds obviously. $\qquad \square$

Now we are ready to prove Theorem 1. The main idea is to compare the terms in $S_J^{\vartheta^*}$ to the increments of the solution over small intervals.

*Proof.* Let $\varphi \in C(\overline{\mathbb{R}})$ be an even function. Assume that $\varphi$ is decreasing on $(0, \infty)$ (the case that $\varphi$ is increasing on $(0, \infty)$ can be proved similarly). For any $h > 0$, define the function $\varphi_h$ by

$$\varphi_h(x) = \int_{-\infty}^{\infty} \frac{1}{\sqrt{2\pi}} \varphi(x + hy) e^{-\frac{y^2}{2}} dy.$$

By the Approximation Lemma (Feller, 1991), we have that

$$\lim_{h \to 0} \sup_{x \in \mathbb{R}} |\varphi(x) - \varphi_h(x)| = 0. \tag{30}$$

It follows from direct calculation that

$$\begin{aligned}
\varphi_h(x) &= \int_{-\infty}^{\infty} \frac{1}{\sqrt{2\pi}} \varphi(x + hy) e^{-\frac{y^2}{2}} dy \\
&= \int_{-\infty}^{\infty} \frac{1}{\sqrt{2\pi}} \varphi(-x - hy) e^{-\frac{y^2}{2}} dy \\
&= \int_{-\infty}^{\infty} \frac{1}{\sqrt{2\pi}} \varphi(-x + hy) e^{-\frac{y^2}{2}} dy \\
&= \varphi_h(-x).
\end{aligned}$$

Thus $\varphi_h$ is also an even function. In addition, we have

$$\begin{aligned}
\dot{\varphi}_h(x) &= \int_{-\infty}^{\infty} \frac{1}{\sqrt{2\pi}h^3} \varphi(x + y) y e^{-\frac{y^2}{2h^2}} dy \\
&= \int_{0}^{\infty} \frac{1}{\sqrt{2\pi}h^3} (\varphi(y + x) - \varphi(y - x)) y e^{-\frac{y^2}{2h^2}} dy.
\end{aligned}$$

Since $\varphi$ is decreasing on $(0, \infty)$, it follows that

$$\mathrm{sgn}\left(\dot{\varphi}_h(x)\right) = -\mathrm{sgn}(x).$$

We continue to use $\{H_t(x)\}_{t \in [0,1]}$ to denote the functions defined in (19) with $\varphi_h$ in place of $\varphi$. Let $\{L_{m,J}(x)\}_{m=1}^{J}$ be functions defined in (24) with $\{H_t(x)\}_{t \in [0,1]}$ here.

For a large enough $J$, let $\vartheta^*$ be the strategy defined in Definition 1, and let $\eta_n \sim \mathcal{S}(\Delta_n, x/\sigma_0)$, by direct calculation we obtain

$$E_P\left[\varphi_h\left(S_J^{\vartheta^*}\right)\right] - E_P[\varphi_h(\sigma_0 \eta_n)]$$

$$= E_P\left[\mathbf{H}_1\left(S_J^{\vartheta^*}\right)\right] - \mathbf{H}_0(x)$$

$$= E_P\left[\mathbf{H}_1\left(S_J^{\vartheta^*}\right)\right] - E_P\left[H_{\frac{J-1}{J}}\left(S_{J-1}^{\vartheta^*}\right)\right] + E_P\left[H_{\frac{J-1}{J}}\left(S_{J-1}^{\vartheta^*}\right)\right] - E_P\left[H_{\frac{J-2}{J}}\left(S_{J-2}^{\vartheta^*}\right)\right] + \cdots$$

$$+ E_P\left[H_{\frac{m}{J}}\left(S_m^{\vartheta^*}\right)\right] - E_P\left[H_{\frac{m-1}{J}}\left(S_{m-1}^{\vartheta^*}\right)\right] + \cdots + E_P\left[H_{\frac{1}{J}}\left(S_1^{\vartheta^*}\right)\right] - \mathbf{H}_0\left(T_0^{\vartheta^*}\right)$$

$$= \sum_{m=1}^{J} \left\{ E_P\left[H_{\frac{m}{J}}\left(S_m^{\vartheta^*}\right)\right] - E_P\left[H_{\frac{m-1}{J}}\left(S_{m-1}^{\vartheta^*}\right)\right] \right\}$$

$$= \sum_{m=1}^{J} \left\{ E_P\left[H_{\frac{m}{J}}\left(S_m^{\vartheta^*}\right)\right] - E_P\left[L_{m,J}\left(S_{m-1}^{\vartheta^*}\right)\right] \right\} + \sum_{m=1}^{J} \left\{ E_P\left[L_{m,J}\left(S_{m-1}^{\vartheta^*}\right)\right] - E_P\left[H_{\frac{m-1}{J}}\left(S_{m-1}^{\vartheta^*}\right)\right] \right\}$$

$$=: I_{1n} + I_{2n}.$$

According to Lemma F.3 and (4) in Lemma F.1, we can infer

$$|I_{1n}| + |I_{2n}| \leq K'' \left( \frac{\log J}{\sqrt{J}} + \frac{1}{\sqrt{J}} \right),$$

where $K''$ is a constant. It implies that

$$\lim_{h \to 0} \left| E_P\left[\varphi_h\left(T_J^{\vartheta^*}\right)\right] - E_P[\varphi_h(\sigma_0 \eta_n)] \right| = O\left( \frac{\log J}{\sqrt{J}} + \frac{1}{\sqrt{J}} \right). \tag{31}$$

Putting together (30) and (31), we have

$$\lim_{J \to \infty} \left| E_P \left[ \varphi \left( S_J^{\vartheta^*} \right) \right] - E_P \left[ \varphi \left( \sigma_0 \eta_n \right) \right] \right|$$

$$\leq \lim_{h \to 0} \lim_{J \to \infty} \left| E_P \left[ \varphi \left( S_J^{\vartheta^*} \right) \right] - E_P \left[ \varphi_h \left( S_J^{\vartheta^*} \right) \right] \right|$$

$$+ \lim_{h \to 0} \lim_{J \to \infty} \left| E_P \left[ \varphi_h \left( T_J^{\vartheta^*} \right) \right] - E_P \left[ \varphi_h \left( \sigma_0 \eta_n \right) \right] \right|$$

$$+ \lim_{h \to 0} \left| E_P \left[ \varphi_h \left( \sigma_0 \eta_n \right) \right] - E_P \left[ \varphi \left( \sigma_0 \eta_n \right) \right] \right|$$

$$=0.$$

Then we complete the proof of Theorem 3.2. Theorem 3.3 is a corollary directly from Theorem 3.2. □