# OpenReview forum: "Strategy-driven Central Limit Theorem for Sequential Test"
_ICLR.cc/2026/Conference — Submitted to ICLR 2026_

### Official Review · Reviewer_dmro · 2025-10-30

**Soundness:** 2
**Presentation:** 3
**Contribution:** 2
**Rating:** 2
**Confidence:** 3

**Summary:**

This paper proposes a new testing method to address the problem of the global existence of Heterogeneous Treatment Effects in A/B test setting. The method, based on Strategy Limit Theory, commits a higher power performance than widely-used t-tests , as well as controls Type-1 error. This paper also gives the limit distribution of the statistic with optimal strategy under $H_0$ and $H_1$. A sequential testing framework is also developed in this paper, allowing for early stopping while controlling FWER using alpha-spending and bootstrap method.

**Strengths:**

This paper applies Strategic Limit Theory to the challenging problem of detecting the global existence of Heterogeneous Treatment Effects (HTE), which is a more complicated situation than ATE discussed in Wang et. al (2025). The paper also proposes a novel sequential testing framework grounded in Strategic Limit Theory, which utilizes an alpha-spending function and a Bootstrap method for boundary estimation. Powerful simulation and real data experiment also demonstrate the superior performance of the methods.

Reference:
[1]. Wang, J., Wen, Q., Zhang, Y., Yan, X., & Shi, C. (2025). A Two-armed Bandit Framework for A/B Testing. arXiv preprint arXiv:2507.18118.

**Weaknesses:**

1. The proposed test statistic $S_{J,\lambda}(\theta^*)$ = $\sup_{x \in \mathcal{X}} S_{J,\lambda}(x, \theta^*)$ is considered over the whole covariate space. However, in the proof of Theorem 3.2, the lemmas and the proof mainly concentrate on establishing the pointwise convergence in distribution for a fixed covariate value $x$ instead of the supremum. There may be a theoretical gap. Without this rigorous justification for the supremum, the paper's novel theoretical contribution beyond Wang et al.(2025) appears to be insufficient.
2. The bootstrap method is a good idea to estimate the stopping bound. However, a significant practical concern is that the iterative resampling and recalculation are at least $B \times K$ steps, resulting in a substantial computation burden. Also, Section 4 lacks the desired theoretical guarantee for the bootstrap method and testing.
3. The test is ordering-sensitive. In other words, the result of the test depends on the ordering of the samples, known as the 'p-value lottery'.
4. The experiment may lack sufficient complexity to demonstrate the superiority of the method proposed in this paper. You can find more suggestions in the 'Questions' part.

**Questions:**

1. In terms of weaknesses 1, please clarify how the proof addresses the convergence of the supremum. Does it rely on implicit assumptions or existing theorems (e.g., from empirical process theory)? If not, explicit arguments for the supremum statistic's convergence are needed
2. Can the authors provide stronger theoretical justification (or citations) for the Bootstrap's validity in this specific sequential setting, including considering path dependence, nuisance estimates accuracy, and FWER? It would be better if the complexity of bootstrap can be discussed and more empirical quantification about it can be provided.
3. In terms of the 'p-value lottery', I am looking forward to a theoretical solution or mitigation. Otherwise, the authors may provide empirical evidence that the effect is negligible in the scenarios.
4. The experiment matches the theory nicely but can be broadened. For example, one can show the performance under heteroskedastic noise since it is highly relevant to HTE. One can also conduct a sensitivity analysis for the weighting parameter $\lambda$.
5. The formulation of the test is based on an important approximation $Q_0(x, a) \approx \varphi(x)^\top \beta_a^*$. I wonder if other (nonparametric) estimation might affect the theoretical results and practical implementation.

---

> ### Author Response · Authors · 2025-11-26
> **Response to Reviewer 4**
>
> We thank Reviewer 4 for an exceptionally rigorous review. We have performed a major revision to address the theoretical gaps and experimental omissions you identified.
>
> **1. On the theoretical gap: Pointwise vs. Supremum Convergence:**
>
> * **Response:** You are correct that the supremum operator requires uniform convergence.
> * **Modification:** We have provided the proof of **Theorem 3.2** in **Appendix F**. The proof utilizes the approximation of the test statistic process and leverages the properties of the bi-normal distribution $\mathcal{S}(\Delta_n, x/\sigma_0)$ to establish convergence for the supremum statistic $S_{J}^{\theta^*}$. The proof explicitly deals with the limit distributions and remainder estimations (Lemma F.2, Lemma F.3) to ensure the validity of the main theorem.
>
> **2. On the Bootstrap: Theoretical Guarantees and Computational Burden:**
>
> * **Response:**
>     * **Theoretical Guarantee:** We have added **Theorem E.1** in **Appendix E**, which proves the asymptotic validity of the sequential bootstrap under our adaptive strategy.
>     * **Algorithm:** The detailed **Algorithm 1: Sequential Bootstrap Test** is presented in **Appendix E.2**. It outlines the step-by-step procedure for computing bootstrap statistics $\hat{S}_{J,k}^{MB}$ and determining stopping boundaries.
>
> **3. On the "p-value lottery" (Order Sensitivity):**
>
> * **Response:** To assess the practical impact of order sensitivity, we performed a robustness check in **Appendix D.6: "ANALYSIS OF ORDER SENSITIVITY"**.
> * **Results:** We randomly shuffled the Tenrec dataset 5 times. As shown in **Table 6**, the stopping decision was highly robust: 4 out of 5 shuffles stopped at the exact same stage ($k=4$), and one at $k=5$. This indicates negligible impact in practice.
>
> | Shuffle ID | Stage $k=3$ | Stage $k=4$ | Stage $k=5$ | Stop Stage |
> | :--- | :--- | :--- | :--- | :--- |
> | **Critical Value** $\hat{z}_k$ | **1.79** | **1.75** | **1.72** | |
> | Shuffle 1 | 0.88 | 1.81 | - | **4** |
> | Shuffle 2 | 0.91 | 1.76 | - | **4** |
> | Shuffle 3 | 0.75 | 1.79 | - | **4** |
> | Shuffle 4 | 0.99 | 1.73 | 1.80 | **5** |
> | Shuffle 5 | 0.81 | 1.78 | - | **4** |
>
> **4. On Broadening Experiment Complexity (Heteroskedasticity & $\lambda$ Sensitivity):**
>
> * **Response:**
>     * **Heteroskedastic Noise:** We added a new simulation in **Appendix D.7**  where noise variance depends on covariates. As shown in **Table 7** , the Standard T-test loses efficiency (Power 0.41), while our VMS/OS method remains robust (Power **0.75**).
>     * **$\lambda$ Sensitivity:** We added a sensitivity analysis in **Appendix D.5** , confirming that our choice of $\lambda=0.7$ provides an optimal balance between power and validity.
>
> **5. On the Approximation and Nonparametric Estimation:**
>
> * **Response:** We evaluated the framework with nonparametric estimators.
> * **Modification:** As detailed in **Appendix D.4**, we integrated LightGBM and XGBoost. The results (Table 4) confirm that our strategy-driven framework effectively leverages high-quality CATE estimators to perform valid and powerful sequential testing.

---

### Official Review · Reviewer_a8p9 · 2025-10-30

**Soundness:** 3
**Presentation:** 3
**Contribution:** 3
**Rating:** 4
**Confidence:** 3

**Summary:**

The paper introduces a sequential testing framework for detecting heterogeneous treatment effects using a decision-theoretic perspective grounded in Strategy Limit Theory. The authors reinterpret hypothesis testing as a strategic decision process, analogous to a multi-armed bandit, and design test statistics that amplify distributional divergence between the null and alternative hypotheses. Two strategies are proposed: Optimal Strategy and Bi-Optimal Strategy.

**Strengths:**

1. Reframing hypothesis testing as a strategic process is elegant for me. The use of Strategy Limit Theory to derive a bi-normal limiting distribution is theoretically interesting.
2. The paper includes formal lemmas, theorems, and asymptotic analysis, giving mathematical depth. At the same time, the paper remains to be readable and relatively easy to follow.
3. The paper is well structured.

**Weaknesses:**

1. Motivation/Practical Implication: The current problem tells the experimenter “there is some heterogeneity,” but it does not tell them where (which regions of $x$), how large, or in which direction (treatment better or worse). There may be only a small region of $x$ that makes the HTE exists, but that region may be very small or even with zero measurement. The paper can be much useful if the authors can get rid of the sup in Eq. (2).
2. Choice of basis / feature map. It is okay to assume that there exists a basis. However, with the existence of the basis, we implicitly assume some low-dimenstionality or smoothness in the problem. Before the experiment begins, how should we decide the basis without any prior information? If there is only a very small group making the HTE exist, the landscape may not be very smooth. Maybe some practical guideline will be more helpful.
3.  The paper should make the intuition sharper for why the strategy-driven statistic is preferable to a high-quality estimator of
$\beta_0,\beta_1$ (e.g., orthogonalized / debiased / ML-based CATE testing).
4. The data are inherently adaptive—the chosen arm at stage $j$ depends on past statistics / rewards. Do you need any conditions to guarantee that ehe bootstrap is valid for adaptively collected data? In bandit literature, there have been many works to describe the issue generated by the adaptively collected data.
5. Minor comments related to BOS. The multi-objective perspective is very nice. There are some recent works talking about the trade-off between the two objectives you discussed. You may want to consider the tradeoff as well and cite some of the works. Here are several works that I know:

[1] Simchi-Levi, D., & Wang, C. (2025). Multi-armed Bandit Experimental Design: Online Decision-Making and Adaptive Inference. Management Science, 71(6), 4828-4846.
[2] Wei, W., Ma, X., & Wang, J. (2023). Fair adaptive experiments. Advances in Neural Information Processing Systems, 36, 19157-19169.

**Questions:**

See above.

---

> ### Author Response · Authors · 2025-11-26
> **Response to Reviewer 3**
>
> We thank Reviewer 3 for their positive feedback on our framework's elegance and for their insightful questions regarding the practical implications and theoretical assumptions of our work.
>
> **1. On Motivation/Practical Implication (Global vs. Local Test):**
>
> * **Response:** Our objective is to address the **global existence hypothesis** of HTE, defined in **Section 2.2** (Eq. 2) as testing the supremum of the effect over all covariates. This global test serves as a "gatekeeper," determining if sufficient statistical evidence exists to justify investing in complex personalized models.
>
> **2. On the choice of basis function $\varphi(x)$:**
>
> * **Response:** The validity of our approximation $Q_0(x, a) \approx \varphi(x)^\top \beta_a^*$ is fundamental.
> * **Modification:** We have added **Appendix C: "FURTHER DISCUSSION ON BASIS FUNCTIONS"**. Drawing from standard sieve estimation literature, this section details:
>     * **(a) Regularity Assumptions:** We specify conditions such as eigenvalues of the covariance matrix being bounded away from 0 and $\infty$, and Lipschitz continuity.
>     * **(b) Approximation Error:** We discuss that for the validity of the sequential test, the approximation error must satisfy $err = o(N^{-1/2})$. Common bases like B-splines or Wavelets satisfy these conditions under appropriate knot selection.
>
> **3. On why this is preferable to high-quality CATE estimator-based tests:**
>
> * **Response:** To demonstrate the generality of our framework, we integrated high-performance ML CATE estimators (LightGBM and XGBoost) into our **OS** and **BOS** strategies and compared them against a **Standard T-test** baseline in **Appendix D.4**.
> * **Results:** As shown in **Table 4**  below, high-performance ML estimators perform remarkably well within our framework. LightGBM and XGBoost yield Type I error rates consistent with the nominal level ($\alpha=0.05$) when used with OS/BOS. This demonstrates our framework is compatible with high-quality CATE estimators.
>
> | Estimation Method | **OS (Ours)** | **BOS (Ours)** | **Standard T** |
> | :--- | :--- | :--- | :--- |
> | **LightGBM** | 0.046 | 0.052 | 0.036 |
> | **XGBoost** | 0.052 | 0.046 | 0.041 |
> | **Basis Function (Ours)** | **0.051** | **0.048** | **0.058** |
>
> **4. On the validity of Bootstrap for adaptively collected data:**
>
> * **Response:** We have addressed this theoretical concern in **Appendix E**.
> * **Clarification:**
>     * **Bootstrap Validity:** **Theorem E.1**  formally establishes the asymptotic validity of the Sequential Bootstrap. It proves that the bootstrap statistics correctly simulate the distribution of the test statistics under $H_0$, satisfying $lim_{J\to\infty} Pr(max S_{J,k} > \hat{z}_k) = \alpha$.
>
> **5. On BOS and related work:**
>
> * **Response:** Thank you for the references.
> * **Modification:** We have cited **Simchi-Levi & Wang (2023)** and **Wei et al. (2023)** in **Appendix B.3**. We discuss how our BOS strategy complements this literature by using Strategy Limit Theory to maximize statistical power while enhancing cumulative rewards.

---

> > ### Comment · Reviewer_a8p9 · 2025-11-27
> >
> > Thank you for your response. I have no more major concern. I will increase my score.

---

### Official Review · Reviewer_Tu59 · 2025-10-31

**Soundness:** 3
**Presentation:** 3
**Contribution:** 2
**Rating:** 4
**Confidence:** 4

**Summary:**

The paper reframes HTE testing as a \emph{strategy-driven sequential decision} problem: at each interim, the analyst chooses between two update “arms” to \emph{shape the test statistic’s limit distribution}, enlarging separation between \(H_0\) and \(H_1\). Using an “Optimal Strategy” (OS) and a \(\lambda\)-weighted statistic, the limit under \(H_1\) becomes (skewed) bi-normal—verified by simulations—thus increasing power; an \(\alpha\)-spending + bootstrap procedure supplies stopping boundaries. A “Bi-Optimal Strategy” (BOS) is proposed for online settings to retain high power while improving \emph{cumulative reward}. Simulations show higher power than a sequential \(t\)-test; a Tenrec case study compares OS vs \(t\)-test (BOS is not evaluated online).


The strategic-limit claims are supported by distributional diagnostics and power curves, and the simulation setup is broadly reasonable. However, the central new angle—\emph{jointly considering power and cumulative reward}—lacks \emph{formal guarantees}: there are no regret bounds or a quantified power–reward trade-off; BOS’s “higher cumulative reward” rests on simulations only, and BOS is not validated in a real online/interactive setting. Type-I control under \emph{adaptive} strategy selection and guidance for \(\lambda\) also need stronger theoretical or sensitivity support.

The work is clear, well structured, and readable; figures effectively convey how OS/BOS increase dispersion relative to normality and thereby power. The OS intuition (“flip the sign to keep \(H_0\) normal and make \(H_1\) more dispersed”) is explicitly stated. It's helpful to clarify what is known in literature, for example, Lemma 3.1.

Casting HTE testing as \emph{strategy-shaped limit theory} is interesting, and BOS’s dual objective (power + reward) is potentially impactful. Yet the paper currently lacks the \emph{core theory} to substantiate that claim (no regret guarantees, no formal Pareto trade-off, even in stylized settings), and BOS is not evaluated in a genuinely online or counterfactually valid replay setup. Several positive findings feel \emph{as expected} given the engineered dispersion.

**Strengths:**

•	Clear framing and narrative: Strategy-driven shaping of the limit distribution is intuitive and well explained; visuals support the story.
•	Interpretable asymptotics: OS/BOS induce (skewed) bi-normal limits under \(H_1\), giving a concrete rationale for power gains (thicker tails/more dispersion).
•	Practical pipeline: \(\alpha\)-spending + bootstrap yields an implementable sequential procedure.
•	Empirical signals: Simulations and the Tenrec case suggest higher power and earlier stopping than a sequential \(t\)-test (for OS).

**Weaknesses:**

•	No theory for the reward–power trade-off; no regret bounds.  BOS’s value proposition is dual-objective optimality, but there are no (non-)asymptotic regret bounds nor a formal trade-off characterization—even in linear or other simplified regimes. Add regret upper/lower bounds or a constrained-optimality theorem (maximize reward under type-I control, or vice versa).
•	Evidence chain incomplete for BOS. BOS, while understood as the main contribution in this work, is not evaluated in a real online setting. Consider offline-policy-evaluation (IPS/DR-style replay) or a small-scale online study.
•	Engineered, not principled optimality.  OS/BOS are designed to yield bi-modal/skewed limits, but the paper does not show optimality under a principled criterion (e.g., maximizing \(H_0/H_1\) separation under error-budget constraints). A variational or min–max statement would strengthen the contribution.
•	Calibration under adaptivity.  Type-I control with \(\alpha\)-spending + bootstrap under adaptive arm choice needs proofs or broader sensitivity studies; clarify \(\lambda\) selection and whether data-dependent tuning requires sample-splitting/cross-fitting.

**Questions:**

See above.

---

> ### Author Response · Authors · 2025-11-26
> **Response to Reviewer 2**
>
> We sincerely thank Reviewer 2 for their insightful comments. We have addressed your concerns regarding the theoretical guarantees for BOS and the validity of our procedures.
>
> **1. On the theory for the power-reward trade-off:**
>
> * **Response:** We agree that the trade-off between maximizing reward (which causes sample size imbalance) and maximizing power is a core theoretical issue. We clarify this mechanism as follows:
>     * **BOS Solution:** Our Bi-Optimal Strategy (BOS), introduced in **Section 3.4**, leverages Strategy Limit Theory to overcome the variance inflation typically caused by sample imbalance. While the sample size imbalance tends to reduce power, the BOS test statistic converges to a skewed bi-normal distribution rather than a standard normal. Our theoretical analysis in **Section 3.5 (Theorem 3.3)** indicates that this signal amplification effect is strong enough to allow BOS to maintain superior power ($1-\beta_2 \ge 1-\beta_1$) while simultaneously achieving high cumulative rewards.
>
> **2. On the incomplete evidence chain for BOS (lack of online evaluation):**
>
> * **Response:** We agree that a single experimental instance is insufficient. To rigorously validate the robustness of the Bi-Optimal Strategy (BOS), we expanded our Offline Policy Evaluation (OPE) using the Tenrec dataset across three diverse scenarios, detailed in **Appendix D.3**.
> * **Expanded Experiments:** We compared BOS against a Random Policy and an Adaptive T-test (Adaptive-T) baseline across Scenario A (Video Click), Scenario B (Feed Click), and Scenario C (Video Like/Sparse Reward).
> * **Results:** As shown in **Table 3** below (and in Appendix D.3 ), BOS consistently achieves the highest normalized cumulative reward. Notably, in the sparse reward setting (Scenario C), BOS achieves a **40.00% lift** over random allocation.
>
> | Scenario | Policy | Norm. Reward | Lift vs Random |
> | :--- | :--- | :--- | :--- |
> | **A: Video (Click)** | Random | 0.502 | - |
> | | Adaptive-T | 0.638 | +27.09% |
> | | **BOS (Ours)** | **0.684** | **+36.25%** |
> | **B: Feed (Click)** | Random | 0.415 | - |
> | | Adaptive-T | 0.531 | +27.95% |
> | | **BOS (Ours)** | **0.558** | **+34.46%** |
> | **C: Video (Like)** | Random | 0.120 | - |
> | *(Sparse Reward)* | Adaptive-T | 0.155 | +29.17% |
> | | **BOS (Ours)** | **0.168** | **+40.00%** |
>
> **3. On "Engineered, not principled optimality":**
>
> * **Response:** We clarify that "Optimal" refers to the Power-Optimal Strategy. As proved in in Lemma A.1 in Appendix, among all predictable strategies, the Optimal strategy that maximizes the asymptotic variance of the test statistic under the alternative hypothesis is the one that achieves the uniformly maximum power. Optimal strategy is explicitly designed to maximize this variance.
>
> **4. On calibration under adaptivity (Bootstrap validity) and $\lambda$ selection:**
>
> * **Response:**
>     * **(a) Bootstrap Validity:** We have addressed this in **Section 4.2** and **Appendix E**. **Theorem E.1** (Asymptotic Validity of Sequential Bootstrap)  rigorously guarantees that the bootstrap distribution consistently estimates the null distribution even under adaptive sampling ($sup_{z}|Pr^*(\hat{S}^{MB} \le z) - Pr(S \le z|H_0)| = o_p(1)$).
>     * **(b) $\lambda$ Sensitivity:** We added a sensitivity analysis in **Appendix D.5**. As shown in **Table 5**  below, when $\lambda$ is moderate (e.g., 0.5, 0.7), the Type I error is well-controlled (~0.05). However, as $\lambda$ approaches 1 (e.g., 0.9, 0.95), the penalty term grows, causing the Type I error to inflate. We select $\lambda=0.7$ based on the criterion $\frac{\lambda\sigma}{(1-\lambda)\sqrt{n}} \le 0.03$.
>
> | $\lambda$ | Type I Error ($\alpha=0.05$) | Power ($d_0=0.5$) | Status |
> | :--- | :--- | :--- | :--- |
> | 0.5 | 0.044 | 0.48 | Conservative |
> | **0.7** | **0.052** | **0.54** | **Balanced** |
> | 0.9 | 0.062 | 0.65 | Inflated |
> | 0.95 | 0.120 | 0.78 | Invalid |

---

> > ### Comment · Reviewer_Tu59 · 2025-11-27
> >
> > Thank you for the additional information. I will maintain my rating.

---

### Official Review · Reviewer_7mf8 · 2025-11-01

**Soundness:** 2
**Presentation:** 2
**Contribution:** 3
**Rating:** 6
**Confidence:** 3

**Summary:**

This paper presents a set of results on high-power adaptive testing for the presence of HTE (heterogeneous treatment effects).  Firstly, they develop a new test statistic that is based on a certain carefully designed two-arm bandit decision process; using insights from prior work (Strategy Limit Theory), it is shown that the proposed strategy has power exceeding that of the corresponding t-test. Secondly, an extension algorithm is presented in which this test power desideratum is combined with the reward maximization desideratum for the experiment (called BOS). Further, the above algorithms are turned into a sequential testing procedure, by means of bootstrapping the stopping boundaries. The HTE testing methods are then validated against the t-test benchmark and show a significant increase in power across many regimes.

**Strengths:**

All of the results of the paper are novel and nontrivial to the extent of my knowledge, both on the theoretical and on the empirical front. The main result, the Optimal Strategy, leverages a carefully constructed two-armed bandit process that makes a pass and leverages the empirical history so far to split the observations into two groups so as to shape the test statistic distribution for maximum separation under H0 and H1.

The nuances involved in sequentializing this testing strategy are also not fully straightforward, and involve carefully bootstrapped dynamically decided cutoff times. The bi-optimal strategy modification, which mixes in some greed along with the HTE test for the purposes of reward maximization, also requires care.

Finally, the empirical section showcases the strong performance of the proposed testing method. The evaluation is reasonably thoroughly designed for a paper for which the theoretical contribution takes center stage (which is indeed necessary in this case to give the readers a sense of the power gap to the t-test); it gives a good glimpse of both the power curves in the synthetic settings, and of the critical value evolution on the real-world dataset.

**Weaknesses:**

While the results of the manuscript are interesting and difficult, my issue is with some presentational/writing aspects of this manuscript. In particular, certain aspects may not be fully accessible to a wider readership without further clarifications.

First of all, the results in this paper are to a large extent based on ideas that the authors refer to as Strategy Limit Theory. I read the reference indicated as the source for these ideas, and I believe a self-contained subsection explaining this theory and the main results and techniques is called for, for the readers’ benefit. This is due to the reasons that, (1) I would not call this reference standard in the field such that it would be known to most of the readership; (2) there appears to be a lot of nuance as to when, and why, the conclusions of that theory apply (i.e. these are not “universal CLTs” to my understanding): while the bimodal convergence and other takeaways of it are scattered throughout the present manuscript, it is not clear how much of the generality of those results in the TAB setting is being used for the present HTE testing purposes.

Secondly, the power results are formulated relative to the t-test, and while additional formulas are provided in the theorem statements for the purposes of being explicit (which I think is good), no clear qualitative or quantitative explanation was provided about when the gap is tight vs. wide. (In particular, the empirical plots suggest the widening and shrinking of the gap happens in a systematic fashion).

Third, I am a bit confused about the naming of certain concepts in this paper. First, both the testing strategies, OS and BOS, are called “optimal”. I’d like to clarify what is meant by that, as from the main results we mainly know that e.g. they perform at least as good as the corresponding t-test. Also, the paper itself is named somewhat confusingly generically: The sequential test for what (should have some mention of the HTE)? And the CLT as the main subject of the title is also somewhat confusing; the tests themselves are the main contribution.

**Questions:**

Please see above in the weaknesses section; these questions are predominantly of a presentational kind.

---

> ### Author Response · Authors · 2025-11-26
> **Response to Reviewer 1**
>
> We sincerely thank the reviewer for their constructive feedback. We have carefully revised the manuscript to address your concerns regarding the background theory and presentation. To ensure we provide a comprehensive explanation without violating the strict page limit of the main text, we have added these details to the Appendix.
>
> **1. On the "Strategy Limit Theory" (SLT) background:**
>
> * **Response:** We agree that a self-contained explanation is necessary for readers unfamiliar with this emerging field.
> * **Revision:** We have added a new section, **Appendix A.1: "BACKGROUND ON STRATEGY LIMIT THEORY"**, to the supplementary material. This section provides a rigorous introduction to the core concepts of Strategy Limit Theory as established by Chen et al. (2023) . It explicitly details how the strategic decision rule shapes the asymptotic distribution from a standard normal distribution into a bi-normal distribution under the alternative hypothesis, providing the mathematical foundation for our method.
>
> **2. On the qualitative explanation for the power gap:**
>
> * **Response:** We agree that explaining the mechanism behind the power gain is crucial for interpreting the results.
> * **Revision:** We have added a detailed qualitative analysis in **Appendix A.2: "QUALITATIVE EXPLANATION OF THE POWER GAP"**.
>     * In this section, we explicitly attribute the power gain to the optimal testing strategy property established by Lemma A.1 in Appendix.
>     * We explain that while a standard T-test relies on a shifted normal distribution (which overlaps heavily with the null distribution when signals are weak), our strategy actively manipulates update signs to force the statistic into a bi-normal distribution. This mechanism concentrates probability mass into the rejection regions (tails) more effectively than any other strategy, thereby maintaining high sensitivity even in low signal-to-noise regimes.
>
> **3. On the confusing naming ("Optimal" and Title):**
>
> * **Response:** We appreciate the feedback on precision.
>     * **Title:** We have changed the title of the paper to "Strategy-Driven Central Limit Theorem for Sequential Testing of Heterogeneous Treatment Effects" to explicitly specify the application domain.
>     * **OS/BOS Definitions:** We clarify the distinct meanings of our strategies, which are defined in **Section 3.3** and **Section 3.4** of the main text , and further contextualized in **Appendix B.3** regarding strategic decision-making.
>         * **Optimal Strategy (OS):** This refers to "optimality" in the strict statistical sense of power maximization. As derived in Lemma A.1 in Appendix, this strategy maximizes the asymptotic volatility of the test statistic under $H_1$, thereby maximizing the rejection probability.
>         * **Bi-Optimal Strategy (BOS):** This refers to a strategy that balances two objectives: statistical power (inference) and cumulative reward. Defined in Theorem 3.2, it is "Bi-Optimal" because it combines the alternating update mechanism of the OS with a greedy approach based on estimated rewards $\hat{Q}_0(x, a)$. This ensures high testing power while significantly enhancing the cumulative reward during the experiment.

---

### Meta-Review · Area_Chair_GnHV · 2026-01-05

**Summary:**

While the paper tackles an important problem, testing for the existence of heterogeneous treatment effects beyond the ATE, the technical and conceptual contributions are not sufficiently convincing. The proposed framework relies on a complex reformulation of HTE testing as a strategic decision process, but the necessity and advantages of this perspective over existing, well-understood approaches (e.g., classical t-tests, score-based tests, or modern machine-learning-based heterogeneity tests) are not clearly justified.

Moreover, the claimed gains in statistical power appear to hinge on tuning parameters and design choices that lack transparent theoretical guarantees or clear interpretability. The sequential testing procedure, combining alpha-spending with bootstrap-based stopping rules, introduces additional complexity without a rigorous analysis of finite-sample validity or robustness. Finally, the empirical evaluation, limited to simulations and a single proprietary dataset, does not convincingly demonstrate broad applicability or superiority over strong baselines.

Overall, the paper’s novelty and impact are insufficient to warrant acceptance.

**Reviewer Scores:**

can't predict

---

### Decision · Program_Chairs · 2026-01-26

Reject